# ReContrast: Domain-Specific Anomaly Detection via Contrastive Reconstruction

Jia Guo[1]    Shuai Lu[2]    Lize Jia[2]    Weihang Zhang[2]    Huiqi Li[1,2] *

[1]School of Information and Electronics, Beijing Institute of Technology, China
[2]School of Medical Technoloy, Beijing Institute of Technology, China
guojia.jeremy@gmail.com, lushuaie@163.com
{3220212096, zhangweihang, huiqili}@bit.edu.cn

## Abstract

Most advanced unsupervised anomaly detection (UAD) methods rely on modeling feature representations of frozen encoder networks pre-trained on large-scale datasets, e.g. ImageNet. However, the features extracted from the encoders that are borrowed from natural image domains coincide little with the features required in the target UAD domain, such as industrial inspection and medical imaging. In this paper, we propose a novel epistemic UAD method, namely ReContrast, which optimizes the entire network to reduce biases towards the pre-trained image domain and orients the network in the target domain. We start with a feature reconstruction approach that detects anomalies from errors. Essentially, the elements of contrastive learning are elegantly embedded in feature reconstruction to prevent the network from training instability, pattern collapse, and identical shortcut, while simultaneously optimizing both the encoder and decoder on the target domain. To demonstrate our transfer ability on various image domains, we conduct extensive experiments across two popular industrial defect detection benchmarks and three medical image UAD tasks, which shows our superiority over current state-of-the-art methods. Code is available at: https://github.com/guojiajeremy/ReContrast

## 1   Introduction

Unsupervised anomaly detection (UAD) aims to recognize and localize anomalies based on the training set that contains only normal images. UAD has a wide range of applications, e.g., industrial defect detection and medical disease screening, addressing the difficulty of collection and labeling of all possible anomalies.

Efforts on UAD attempt to learn the distribution of available normal samples. Most state-of-the-arts utilize networks pre-trained on large-scale datasets, e.g. ImageNet [1], for extracting discriminative and informative feature representations. **Feature reconstruction** [2; 3; 4] and **feature distillation** methods [5; 6] are proposed to reconstruct features of pre-trained encoders as alternatives to **pixel reconstruction** [7; 8], because learned features provide more informative representations than raw pixels [2; 3; 9]. The above approaches can be categorized as epistemic methods, under the hypothesis that the networks trained on normal images can only construct normal regions, but fail for unseen anomalous regions. **Feature memory & modeling** methods [10; 11; 12; 13; 14] memorize and model all anomaly-free features extracted from pre-trained networks in training, and compare them with test features during inference. In feature reconstruction, the parameters of encoders should be frozen to prevent the networks from converging to a trivial solution, i.e., pattern collapse [2; 3]. In feature distillation and feature memory & modeling methods, there are no gradients for optimizing

---

*Corresponding Author

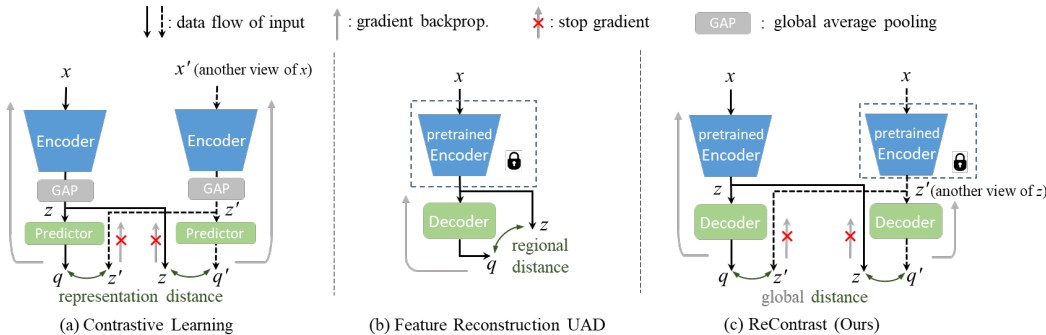

Figure 1: Comparison of architectures. (a) Contrastive learning [16; 17]. (b) Feature reconstruction UAD [2; 9]. (c) Proposed ReContrast. Two decoders share the same weights. z is the output of encoder. q is the output of predictor or decoder. z' and q' is another view of z and q, respectively.

feature encoders. Unfortunately, poor transfer ability is caused by the semantic gap between natural images on which the frozen networks were pre-trained and various UAD image modalities.

Most recently, sparse studies have addressed the transfer problem in UAD settings. CFA [14] and SimpleNet[15] unanimously utilize a learnable linear layer to adapt the output features of the encoder to task-oriented features, while still keeping the encoder frozen. However, the layer after a frozen encoder may be insufficient for adaptation as it can hardly recover the domain-specific information already lost during extraction, especially when the target domain is far from natural images, e.g. medical images. More related works are presented in Appendix A.

In this work, we focus on orienting the whole UAD network in target domains. We propose **Re**constructive **Contrast**ive Learning (ReContrast) for domain-specific anomaly detection by optimizing all parameters end-to-end. We start with epistemic UAD methods that detect anomalies by reconstruction errors between the encoder and the reconstruction network (decoder) [2; 9]. By inspecting the behavior of encoder, we find that direct optimization of the encoder causes diversity degradation of encoder features and thus harms the performance. Accordingly, we introduce **three** key elements of contrastive learning (Figure 1(a)) into feature reconstruction UAD (Figure 1(b)), building a 2-D feature map contrast paradigm to train the encoder and decoder jointly on target images without obstacles. First, to mimic the role of global average pooling (GAP), a new optimization objective, namely global cosine distance, is proposed to make the contrast in a global manner to stabilize the optimization. Second, we bring in the essential operation, i.e., stop-gradient, which prevents positive-pair contrastive learning [16; 17] from pattern collapse. Third, as image augmentation can introduce potential anomalies, we propose to utilize two encoders in different domains to generate representations in two views from the same input image. In addition, we propose a hard-mining strategy for optimizing normal regions that are hard to reconstruct, to magnify the difference between intrinsic reconstruction error and epistemic reconstruction error. It is noted that ReContrast is **NOT** a self-supervised learning method (SSL) for pre-text pre-training like SimSiam [16] and SimCLR [18], but an end-to-end anomaly detection approach.

To validate our transfer ability on various domains, we evaluate ReContrast with extensive experiments on two widely used industrial defect detection benchmarks, i.e., MVTec AD [19] and VisA [20], and three medical image benchmarks, including optical coherence tomography (OCT) [21], color fundus image [22], and skin lesion image [23]. Without elaborately designing memory banks or handcrafted pseudo anomalies, our proposed method achieves superior performance compared with previous state-of-the-arts. Notably, we present an excellent image-level AUROC of 99.5% on MVTec AD, which reduces the error of the previous best by nearly half. The contributions of this study can be summarized as:

- We propose a simple yet effective UAD method to handle the poor transfer ability of the frozen ImageNet encoder when applied to UAD images with a large domain gap. Targeting the similarity and difference between contrastive learning and feature reconstruction, **three** key elements are elegantly introduced into feature reconstruction UAD for optimizing the entire network without obstacles.

- Inspired by the GAP in contrastive learning, we design a new objective function, introducing globality into point-by-point cosine distance for improving training stability.

- We utilize the key operation of positive-pair contrastive learning, i.e., stop-gradient, to prevent the encoder from pattern collapse.

- We propose a novel way of generating contrastive pairs to avoid potential anomalies introduced by image augmentation.

## 2   Method: From Reconstruction to ReContrast

In the following subsections, we present an exploration journey starting from a conventional feature reconstruction baseline to our ReContrast. We propose three cohesive components inspired by the elements of contrastive representation learning and present them in a "motivation, connection, mechanism" manner. We believe such progressive exploration would illustrate the motivation of our method better.

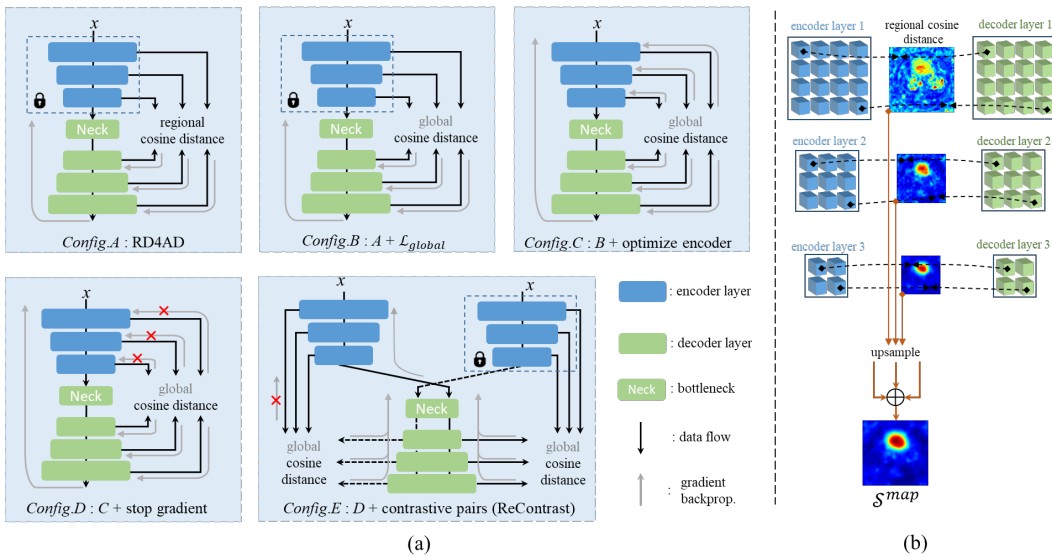

Figure 2: From RD4AD to ReContrast. (a) Training configurations. (b) Calculation of anomaly map. The calculation and optimization of regional and global cosine distance is presented in Figure 3.

### 2.1   Preliminary: Reverse Distillation

First, we revisit a simple yet powerful epistemic UAD method based on feature reconstruction, i.e., Reverse Distillation (RD4AD) [2]. The method is illustrated in *Config.A* of Figure 2. RD4AD consists of an encoder (teacher), a bottleneck, and a reconstruction decoder (student). Without loss of generality, the first three layers of a standard 4-layer network are used as the encoder. The encoder extracts informative feature maps with different spatial sizes. The bottleneck is similar to the fourth layer of the encoder but takes the feature maps of the first three layers as the input. The decoder is the reverse of the encoder, i.e., the down-sampling operation at the beginning of each layer is replaced by up-sampling. During training, the decoder learns to reconstruct the output of the encoder layers by maximizing the similarity between feature maps. During inference, the decoder is expected to reconstruct normal regions of feature maps but fails for anomalous regions as it has never seen such samples.

Let $f_E^k, f_D^k \in \mathbb{R}^{C^k \times H^k \times W^k}$ denote the output feature maps from the $k^{th}$ layer of the encoder and decoder respectively, where $C^k$, $H^k$, and $W^k$ denote the number of channels, height, and width of the $k^{th}$ layer, respectively. Regional cosine distance [2; 6; 24] is used to measure the difference between $f_E^k$ and $f_D^k$ point by point, obtaining a 2-D distance map $\mathcal{M}^k \in \mathbb{R}^{H^k \times W^k}$:

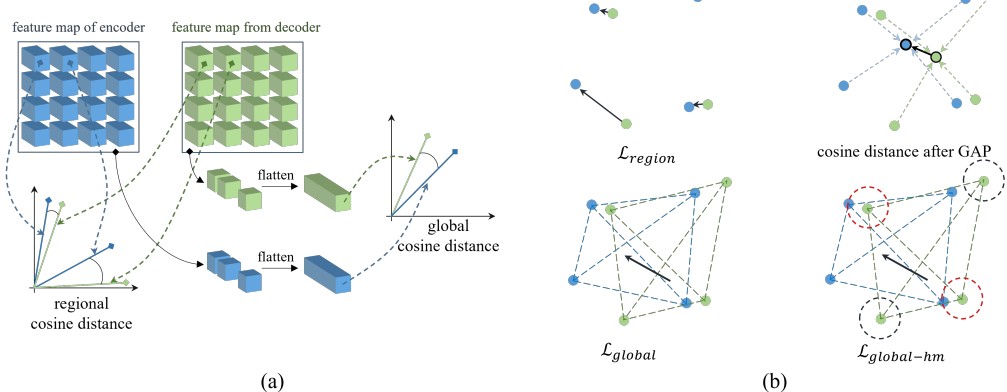

(a)

(b)

Figure 3: Cosine distances. (a) Calculation of regional and global cosine distance. (b) Optimization of distance losses. The black arrows denote optimization direction. The black and red dotted circles indicate the range of whether the feature point pair is close enough to be selected as easy-normal.

$$\mathcal{M}^k (h, w) = 1 - \frac{f_E^k (h, w)^T \cdot f_D^k (h, w)}{\left\| f_E^k (h, w) \right\| \ \left\| f_D^k (h, w) \right\|}, \tag{1}$$

where $\|\cdot\|$ is $\ell_2$ norm, and a set of $(h, w)$ locates a spatial point in the feature map. The calculation is depicted in Figure 3(a). A large value in $\mathcal{M}^k$ indicates anomaly at the corresponding position. Final anomaly score map $\mathcal{S}^{map}$ is obtained by up-sampling all $\mathcal{M}^k$ to image size and a sum operation. The maximum value of $\mathcal{S}^{map}$ is taken as the anomaly score for the image, denoted as $\mathcal{S}^{img}$. During training, the anomaly maps are minimized on normal images by the regional cosine distance loss, given as:

$$\mathcal{L}_{region} = \sum_{k=1}^{3} \frac{1}{H^k W^k} \sum_{h=1}^{H^k} \sum_{w=1}^{W^k} 1 - \frac{f_E^k (h, w)^T \cdot f_D^k (h, w)}{\left\| f_E^k (h, w) \right\| \ \left\| f_D^k (h, w) \right\|}. \tag{2}$$

The optimization of $\mathcal{L}_{region}$ is illustrated in the upper-left of Figure 3(b). Notably, though $\mathcal{L}_{region}$ was proposed clearly as the loss function of RD4AD [2], the official code[1] implemented a **different** function that makes an extra dimension flatten operation (maybe by coding bug). However, we observe extreme instability during the training with $\mathcal{L}_{region}$, producing degenerated I-AUROC of 95.3% on MVTec AD compared with 98.5% reported in the paper. The I-AUROC of 95.3% is more consistent with other methods using $\mathcal{L}_{region}$ [6; 24]. In the next subsection, we will show that the seemingly unreasonable "bug" in the code improves the performance by explicitly introducing globality into optimization.

## 2.2 Global Cosine Similarity

It is a convention to use global average pooling (GAP) to pool the 2-D feature map of the last network layer to a 1-D representation, in both supervised classification and self-supervised contrastive learning. In UAD, GAP will simply mess up all feature points together, losing the ability to distinguish normal and anomalous regions, as shown in the upper-right of Figure 3(b). Therefore, we try to directly bring globality into the loss function while maintaining the point-to-point correspondence between $f_E^k$ and $f_D^k$. Let $\mathcal{F}$ denote a flatten operation that casts a 2-D feature map $f \in \mathbb{R}^{C \times H \times W}$ to a vector $v \in \mathbb{R}^{CHW}$. Our proposed global cosine distance loss is given as:

$$\mathcal{L}_{global} = \sum_{k=1}^{3} 1 - \frac{\mathcal{F} \left( f_E^k \right)^T \cdot \mathcal{F} \left( f_D^k \right)}{\left\| \mathcal{F} \left( f_E^k \right) \right\| \ \left\| \mathcal{F} \left( f_D^k \right) \right\|}, \tag{3}$$

[1]https://github.com/hq-deng/RD4AD

where the number of cosine dimension is $C \cdot H \cdot W$ instead of $C$ in $\mathcal{L}_{region}$, as depicted in Figure 3(a). Intuitively, $\mathcal{M}^k$ is also minimized along with the minimization of $\mathcal{L}_{global}$. It is easy to prove that minimizing $\mathcal{L}_{global}$ to 0 equals to minimizing $\mathcal{M}^k$ to 0. Though losses are not completely optimized to 0 in neural network training, the regional $\mathcal{S}^{map}$ still works as the anomaly map. The model trained with $\mathcal{L}_{global}$ is named as *Config.B*.

As shown in Figure 5(a), the performance of *Config.B* (blue) is much more favorable and stable than *Config.A* (purple) during training. Here, we briefly analyze the underlying mechanism by plotting the landscape of $\mathcal{S}^{map}$ (average of all samples) against model parameters near optimized minima in Figure 4. We observe two differences. First, the overall landscape of the model optimized by $\mathcal{L}_{global}$ is flatter than that by $\mathcal{L}_{region}$, which implies $\mathcal{S}^{map}$ is more stable during training with $\mathcal{L}_{global}$. The instability of $\mathcal{L}_{region}$ can be caused by the point-by-point distance, where the fitting of one region may cause under-fitting of the others. $\mathcal{L}_{global}$ can be regarded as the distance between manifolds of feature points, as depicted in the lower-left of Figure 3(b), which measures the consistency globally instead of excessively focusing on individual regions. Second, the landscape around the final minima of the model trained by $\mathcal{L}_{global}$ is sharper than that by $\mathcal{L}_{region}$. Previous explorations on the geometry of loss landscapes suggest that flat minima generalize better on unseen samples [25; 26], which is unwanted in UAD settings. The real mechanism can be more complicated and is still under further investigation.

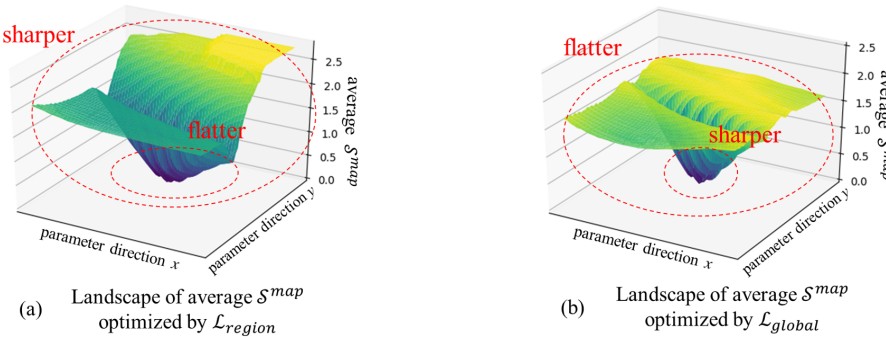

(a) Landscape of average $\mathcal{S}^{map}$ optimized by $\mathcal{L}_{region}$

(b) Landscape of average $\mathcal{S}^{map}$ optimized by $\mathcal{L}_{global}$

Figure 4: $\mathcal{S}^{map}$ landscape in parameter space (reduced to 2-dimension), on APTOS [22], visualized using [27].

## 2.3   Feature Degradation of Optimized Encoder

It seems easy to adapt the network in the target domain by jointly optimizing the encoder and decoder, as illustrated in Figure 2 *Config.C*. However, previous attempts [9] suggest that training encoders would lead encoders to extract indiscriminative features that are easy to reconstruct, which is also described as pattern collapsing or converging to trivial solutions [2]. We observe a similar phenomenon that the training loss quickly decreases to nearly zero, as shown by the orange dotted line in Figure 5(a).

To inspect the behavior of the encoder, we probe the diversity of encoder feature maps by the standard deviation (*std*) of $\ell_2$ normalized feature points $f_E^k(h, w) / \left\| f_E^k(h, w) \right\|_2$. The larger the *std*, the more discriminative the feature, and the more distinctive between different regions. We plot the feature diversity of outputs of the $2^{nd}$ encoder layer during training in Figure 5(b). The *std* of *Config.C* drops quickly during training, while the *std* of *Config.B* keeps constant as a comparison. In addition, the *std* of *Config.C* does not completely collapse to zero, which explains the passable performance. The outputs of other middle layers follow the same trend. The observation indicates that the degeneration of performance is attributed to the degradation of encoder features. It is noted that the *std* starts from different values because of the different modes of batch normalization (BN) [28] during training.[2]

---

[2]Encoder BN is in *eval* mode in *Config.A* using pre-trained running mean and variation, and it is in *train* mode using batch mean and variation if optimized.

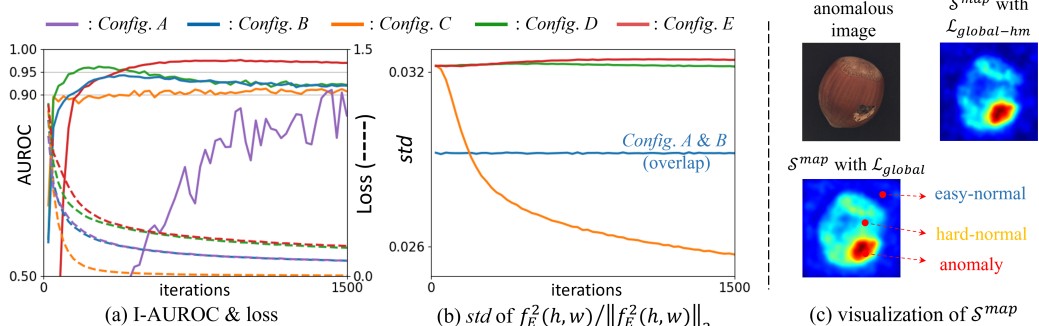

Figure 5: (a) I-AUROC and loss during training on APTOS[22]. (b) Feature diversity during training on APTOS[22]. (c) Example $\mathcal{S}^{map}$ of a *Hazelnut*, optimized by $\mathcal{L}_{global}$ or $\mathcal{L}_{global-hm}$.

## 2.4 Stop Gradient

Contrastive learning for SSL maximizes the similarity between the representations of two augmentations (views) of one image while trying to avoid collapsing solutions in which all inputs are encoded to a constant value. SimSiam [16] indicated that the stop-gradient operation plays an essential role in preventing collapsing for SSL methods with positive pairs only [16; 17]. Intriguingly, we found that contrastive SSL can be explained from the perspective of feature reconstruction. The predictor (a multi-layer perceptron, MLP) can be regarded as a reconstruction network that constructs the representation of one view $x$ to match the representation of another view $x'$, as in Figure 1. Vice versa, feature reconstruction UAD can be transformed into contrastive learning as well, regarding the reconstruction network as a predictor.

To this end, we introduce the stop-gradient operation into feature reconstruction, transforming it into a contrastive-like paradigm. The network is optimized by contrasting the feature maps of the encoder and decoder, as shown in *Config.D* of Figure 2. The gradients do not propagate directly into the encoder, but back into the encoder through the decoder. It is implemented by modifying (3) as:

$$\mathcal{L}_{global} = \sum_{k=1}^{3} 1 - \frac{sg\left(\mathcal{F}\left(f_E^k\right)\right)^T \cdot \mathcal{F}\left(f_D^k\right)}{sg\left(\left\|\mathcal{F}\left(f_E^k\right)\right\|\right) \ \left\|\mathcal{F}\left(f_D^k\right)\right\|} \tag{4}$$

where *sg* denotes the stop-gradient operation. The training of this configuration can be intuitively explained as a "mutual reinforcement". The optimization of $f_D^k$ drives the encoder to be more specific and informative in the target domain by the gradients from the decoder, while the more domain-specific $f_E^k$ of encoder requires further optimization of $f_D^k$. There are previous works trying to explain the deeper mechanism of stop gradient in contrastive learning. In [16], the authors hypothesize this configuration as an implementation of an Expectation-Maximization (EM) algorithm that implicitly solves two underlying sub-problems with two sets of variables. In [29], the authors argue that there are flaws in the hypothesis of [16] and suggest that the decomposed gradient vector (center gradient) helps prevent collapse via the de-centering effect.

As shown in Figure 5(a) and (b), *Config.D* can constantly extract more discriminative features without degeneration, boosting it to perform better than *Config.C* at the beginning of training. Previous works [9; 3] suggested that the reconstruction decoder may learn an "identical shortcut" that well reconstructs both seen and unseen samples, which explains the performance drop of *Config.D* after reaching the peak in Figure 5(a). To handle this issue, contrastive pairs are introduced.

## 2.5 Contrastive Pairs

Contrastive learning methods make use of (positive) contrastive image pairs, based on the intuition that the semantic information of different augmented views of the same image should be the same. Without image augmentations ($x=x'$), contrastive learning degrades into a self-reconstruction paradigm that the predictor predicts the input of itself, which may also collapse into an "identical shortcut" just like *Config.D*. In epistemic UAD methods, it is impossible to construct contrastive pairs by image augmentation, because 1) any augmentation (flip, cutout, color jitter, etc.) could be potential

anomalies; and 2) the feature maps of two augmented views lost the regional correspondence after spatial augmentations (shift, rotate, scale, etc.). Therefore, we propose an augmentation-free method to construct two views of one image.

In previous explorations, the encoder is adapted to the target image domain and generates feature representations in a domain-specific view. We additionally introduce a frozen encoder network without any adaptation throughout the training, which perceives images in the view of pre-train image domain. As shown in *Config.E* of Figure 2, the decoder and bottleneck take the features of the domain-specific encoder to reconstruct the frozen encoder; vice versa, they simultaneously take the features of the frozen encoder to reconstruct the domain-specific encoder, building a complete two-view contrastive paradigm. This configuration can also be interpreted as cross-reconstruction, as the decoder reconstructs the information of the frozen encoder given the input of the adapted encoder; and reconstructs the information of the adapted encoder given the input of the frozen encoder. With six pairs of feature maps, the $\mathcal{S}^{map}$ is obtained by adding up the six up-sampled cosine distance maps. As shown in Figure 5(b), the encoder feature diversity of *Config.E* not only holds but slightly increases during training.

### 2.6 Magnify Epistemic Error by Hard-Normal Mining

Due to network information capacity, intrinsic reconstruction errors exist in available normal regions. The errors of normal details and edges (hard-normal) are generally higher than other plain regions (easy-normal), leading to confusion between the intrinsic error of hard-normal regions and the epistemic error of unseen anomalies, as demonstrated in the lower-left of Figure 5(c). It was proposed to hard-mine the hard-normal regions by simply discarding the "easy" pairs of $f_E^k(h,w)$, $f_D^k(h,w)$ with small cosine distance in $\mathcal{L}_{region}$ [24]. However, arbitrarily discarding feature points in $f_D^k$ breaks the global feature point manifolds in $\mathcal{L}_{global}$. Therefore, we propose a hard mining strategy, namely $\mathcal{L}_{global-hm}$, focusing on the optimization of hard-normal regions to widen the gap between the epistemic errors caused by unseen anomalies and the errors of all normal regions. The gradients of easy $f_D^k(h,w)$ are discarded instead of eliminating $f_D^k(h,w)$ itself, by modifying $f_D^k$ as:

$$f_D^k(h,w) = \begin{cases} sg\left(f_D^k(h,w)\right), \; if \; \mathcal{M}^k(h,w) < \mu\left(\mathcal{M}^k(h,w)\right) + \alpha * \sigma\left(\mathcal{M}^k(h,w)\right) \\ f_D^k(h,w), \; else \end{cases}, \quad (5)$$

where $\mu\left(\mathcal{M}^k(h,w)\right)$ is the average regional distance on the dataset (approximated on mini-batch), $\sigma\left(\mathcal{M}^k(h,w)\right)$ is the standard deviation of the regional distance, and $\alpha$ is a hyper-parameter to control the discarding rate. The optimization of $\mathcal{L}_{global-hm}$ is depicted in the lower-right of Figure 3(b). As shown in Figure 5(c), the model trained by $\mathcal{L}_{global-hm}$ activates fewer normal regions than the model trained by $\mathcal{L}_{global}$. The usage of hard mining strategy can also mitigate the over-fitting of easy training samples.

## 3 Experiments

In this section, we evaluate the proposed method on two popular UAD applications, i.e., industrial defect inspection and medical disease screening. We refer to our complete approach (ReContrast+$\mathcal{L}_{global-hm}$) as ReContrast and Ours in the Experiments section for simplicity.

### 3.1 Experimental Settings

**Datasets**. **MVTec AD** is the most widely used industrial defect detection dataset, containing 15 categories of sub-datasets (5 textures and 10 objects). **VisA** is a challenging industrial defect detection dataset, containing 12 categories of sub-datasets. **OCT2017** is an optical coherence tomography dataset [21]. **APTOS** is a color fundus image dataset, available as the training set of the 2019 APTOS blindness detection challenge [22]. **ISIC2018** is a skin disease dataset, available as task 3 of ISIC2018 challenge [23]. Detailed information is presented in Appendix B.

**Implementation**. WideResNet50 [30] pre-trained on ImageNet [1] is utilized as the encoder by default. AdamW optimizer [31] is utilized with $\beta$=(0.9,0.999) and weight decay=1e-5. The learning rates of new (decoder and bottleneck) and pre-trained (encoder) parameters are 2e-3 and 1e-5, respectively. The network is trained for 3,000 iterations on VisA, 2,000 on MVTec AD and ISIC2018, and 1,000 on APTOS and OCT2017. The $\alpha$ in equation (5) linearly rises from -3 to 1 in the first

Table 1: Anomaly detection performance on MVTec AD, measured in I-AUROC (%).

|  | ADTR [3] | RD4AD [2] | CFlowAD [34] | CutPaste [35] | DRAEM [36] | PaDiM [10] | PatchCore [11] | Ours |
|---|---|---|---|---|---|---|---|---|
| Carpet | 98.7 | 98.9 | 98.7 | 93.9 | 97.0 | **99.8** | 98.7 | **99.8** |
| Grid | 95.0 | **100** | 99.6 | 100 | 99.9 | 96.7 | 98.2 | **100** |
| Leather | 98.1 | **100** | **100** | **100** | **100** | **100** | **100** | **100** |
| Tile | 93.8 | 99.3 | 99.9 | 94.6 | 99.6 | 98.1 | 98.7 | **99.8** |
| Wood | 91.2 | **99.2** | 99.1 | 99.1 | 99.1 | **99.2** | **99.2** | 99.0 |
| *Text. Avg.* | 95.4 | 99.5 | 99.5 | 97.5 | 99.1 | 98.9 | 99.0 | **99.7** |
| Bottle | 98.0 | **100** | **100** | 98.2 | 99.2 | 99.9 | **100** | **100** |
| Cable | 96.8 | 95 | 97.6 | 81.2 | 91.8 | 92.7 | 99.5 | **99.8** |
| Capsule | **99.1** | 96.3 | 97.7 | 98.2 | 98.5 | 91.3 | 98.1 | 97.7 |
| Hazelnut | 98.6 | 99.9 | **100** | 98.3 | **100** | 92.0 | **100** | **100** |
| MetalNut | 97.0 | **100** | 99.3 | 99.9 | 98.7 | 98.7 | **100** | **100** |
| Pill | 98.3 | 96.6 | 96.8 | 94.9 | **98.9** | 93.3 | 96.6 | 98.6 |
| Screw | **99.3** | 97.0 | 91.9 | 88.7 | 93.9 | 85.8 | 98.1 | 98.0 |
| Toothbrush | 98.5 | 99.5 | **100** | 99.4 | **100** | 96.1 | **100** | **100** |
| Transistor | 97.9 | 96.7 | 95.2 | 96.1 | 93.1 | 97.4 | **100** | 99.7 |
| Zipper | 97.2 | 98.5 | 98.5 | 99.9 | **100** | 90.3 | 99.4 | 99.5 |
| *Obj. Avg.* | 98.1 | 98.0 | 97.7 | 95.5 | 97.4 | 93.8 | 99.2 | **99.3** |
| *All Avg.* | 97.2 | 98.5 | 98.3 | 96.2 | 98.0 | 95.4 | 99.1 | **99.5** |

Table 2: Anomaly segmentation performance on MVTec AD, measured in P-AUROC/AUPRO (%).

|  | RD4AD [2] | DRAEM [36] | SPADE [37] | PaDiM [10] | PatchCore [11] | Ours |
|---|---|---|---|---|---|---|
| *Text. Avg.* | 97.7 / 95.0 | 97.5 / 92.1 | 92.9 / 88.4 | 96.9 / 93.1 | 97.5 / 93.6 | **98.0 / 96.2** |
| *Obj. Avg.* | 97.9 / 93.4 | 98.3 / 93.3 | 97.6 / 93.4 | 97.8 / 91.6 | 98.4 / 93.3 | **98.6 / 94.8** |
| *All Avg.* | 97.8 / 93.9 | 97.8 / 92.5 | 96.0 / 91.7 | 97.5 / 92.1 | 98.1 / 93.4 | **98.4 / 95.2** |

Table 3: Anomaly detection and segmentation performance on VisA (%).

|  | RD4AD [2] | DRAEM [36] | SPADE [37] | PaDiM [10] | PatchCore [11] | Ours |
|---|---|---|---|---|---|---|
| I-AUROC *Avg.* | 96.0 | 88.7 | 92.1 | 89.1 | 95.1 | **97.5** |
| P-AUROC *Avg.* | 90.1 | 93.5 | 85.6 | 98.1 | **98.8** | 98.2 |
| AUPRO *Avg.* | 70.9 | 72.4 | 65.9 | 85.9 | 91.2 | **92.6** |

one-tenth iterations and keeps 1 for the rest training. More experimental details and environments are presented in Appendix C.

**Metrics**. Image-level anomaly detection performance is measured by the Area Under the Receiver Operator Curve (I-AUROC). F1-score (F1) and accuracy (ACC) are adopted for medical datasets following [32]. The operating threshold of F1 and ACC is determined by the optimal value of F1. For anomaly segmentation, pixel-level AUROC (P-AUROC) and Area Under the Per-Region-Overlap (AUPRO) [33] are used as the evaluation metrics. AUPRO is more meaningful than P-AUROC because of the unbalanced amount of normal and anomalous pixels [20].

## 3.2 Anomaly Detection and Segmentation on Industrial Images

Anomaly detection results on MVTec AD are shown in Table 1. Our approach achieves a superior average I-AUROC of **99.5%**. In the aspect of error ($100\% - \text{I-AUROC}$), we achieve only 0.5%, reducing the previous SOTA error of PatchCore (0.9%) by relatively **44%**. Anomaly segmentation results are shown in Table 2 (average of 15 categories). Our ReContrast achieves SOTA performance measured by both P-AUROC and AUPRO. In AUPRO, we produce **95.2%**, exceeding the previous SOTA RD4AD by 1.3%. We also present SOTA performances on texture (*Text. Avg.*) and object (*Obj. Avg.*) categories, respectively.

Table 4: Anomaly detection performances under unified multi-class setting on MVTec AD and VisA, measured in I-AUROC(%).

| MVTec AD | RD4AD [2] | UniAD [9] | Ours | VisA | RD4AD [2] | UniAD [9] | Ours |
|---|---|---|---|---|---|---|---|
| Carpet | 98.3 | **99.8** | 98.3 | candle | 93.6 | 94.6 | **96.3** |
| Grid | **99.0** | 98.2 | 98.9 | capsules | 67.8 | 74.3 | **77.7** |
| Leather | **100** | **100** | **100** | cashew | **94.6** | 92.6 | 94.5 |
| Tile | 98.1 | 99.3 | **99.5** | chewingum | 95.3 | 98.7 | **98.6** |
| Wood | 99.3 | 98.6 | **99.7** | fryum | 94.4 | 90.2 | **97.3** |
| Bottle | 79.9 | 99.7 | **100** | macaroni1 | 97.2 | 91.5 | **97.6** |
| Cable | 86.8 | 95.2 | **95.6** | macaroni2 | 86.2 | 83.6 | **89.5** |
| Capsule | 96.3 | 86.9 | **97.3** | pcb1 | **96.5** | 94.3 | **96.5** |
| Hazelnut | **100** | 99.8 | **100** | pcb2 | 93.0 | 92.5 | **96.8** |
| MetalNut | 99.8 | 99.2 | **100** | pcb3 | 94.5 | 89.8 | **96.8** |
| Pill | 92.8 | 93.7 | **96.3** | pcb4 | **100** | 99.3 | 99.9 |
| Screw | 96.5 | 87.5 | **97.2** | pip_fryum | **99.7** | 97.3 | 99.3 |
| Toothbrush | **97.5** | 94.2 | 96.7 | | | | |
| Transistor | 93.3 | **99.8** | 94.5 | | | | |
| Zipper | 98.6 | 95.8 | **99.4** | | | | |
| *All Avg.* | 95.8 | 96.5 | **98.2** | *All Avg.* | 92.7 | 91.5 | **95.1** |

Anomaly detection and segmentation results on VisA are shown in Table 3 (average of 12 categories). Our approach achieves a superior average I-AUROC of **97.5%**, exceeding previous SOTA RD4AD by 1.5%. In AUPRO, we produce **92.6%**, outperforming previous SOTA PatchCore by 1.4%.

### 3.3 Multi-Class Anomaly Detection with A Unified Model

Conventional UAD settings train separate models for different object categories. In UniAD [9], the authors proposed to train a unified model for multiple classes. We evaluate our ReContrast under the unified setting on MVTec AD and VisA. Our model is trained for 5,000 iterations on all images containing all 15 MVTec AD categories (or 12 categories in VisA) and evaluated on each category following [9]. The results are presented in Table 4. On MVTec AD, our method produces an I-AUROC of **98.2%**, exceeding the previous SOTA 96.5% of UniAD by a large **1.7%**. On VisA, both our ReContrast (**95.1%**) and baseline RD4AD (92.7%) outperform the previous multi-class SOTA UniAD (91.5%), while the proposed method exceeds RD4AD by 2.4%.

### 3.4 Anomaly Detection on Medical Images

The experimental results on three medical image datasets are presented in Table 5. Apart from industrial UAD methods, we also compare our ReContrast with approaches designed for medical images (f-AnoGan [7], GANomaly [8], AE-flow [32]), and contemporaneous methods proposed for better domain adaption ability (CFA [14], SimpleNet [15]). We reproduce the methods that were not evaluated on the corresponding dataset if they are open-sourced. Our method achieves the best results in all metrics on APTOS and ISIC2018, and comparable best results on the relatively easy OCT2017, demonstrating that ReContrast can directly adapt to various medical image modalities.

### 3.5 Ablation Study

To verify the effect of the proposed elements, we conduct thorough ablation studies on MVTec AD and APTOS. Because some configurations are unstable during training on some categories, we report the last iteration's performances and those of the best checkpoint during training (best by I-AUROC, evaluate every 250 iterations). The results are reported in Table 6. The use of $\mathcal{L}_{global}$ (*Config.B*) boosts the performance of *Config.A* trained by $\mathcal{L}_{region}$. Directly optimizing the encoder and decoder together (*Config.C*) causes a great degeneration because of pattern collapse, while either stop-gradient (*Config.D*) or contrastive pairs (*Config.C+cp*) recovers some of it. By introducing both (*Config.E*), the performance is further improved to exceed the frozen encoder baseline. Finally, training with $\mathcal{L}_{global-hm}$ yields SOTA performances (Ours). Meanwhile, $\mathcal{L}_{global-hm}$ can also improve the baseline reconstruction method alone (*Config.B+hm*). More ablation studies, experimental results with error bars, limitations, and qualitative visualization are presented in Appendix.

Table 5: Anomaly detection performances on medical image datasets (%).

| | APTOS | | | OCT2017 | | | ISIC2018 | | |
|---|---|---|---|---|---|---|---|---|---|
| | I-AUROC | F1 | ACC | I-AUROC | F1 | ACC | I-AUROC | F1 | ACC |
| f-AnoGAN[7] | 89.64 | 89.71 | 85.23 | 83.02 | 88.64 | 82.13 | 79.80 | 67.03 | 68.39 |
| GANomaly[8] | 83.72 | 85.85 | 79.37 | 90.52 | 91.09 | 86.16 | 72.54 | 60.95 | 56.99 |
| PaDiM[10] | 77.17 | 85.47 | 77.24 | 96.88 | 95.24 | 92.80 | 82.13 | 72.04 | 73.06 |
| RD4AD[2] | 92.43 | 90.65 | 86.44 | 99.25 | 97.79 | 96.70 | 85.09 | 74.53 | 78.76 |
| PatchCore[11] | 90.45 | 90.18 | 85.57 | **99.61** | 98.34 | 97.50 | 78.94 | 68.57 | 71.50 |
| AE-flow[32] | N/A | N/A | N/A | 98.15 | 96.36 | 94.42 | 87.79 | 80.56 | 84.97 |
| CFA[14] | 94.21 | 94.39 | 92.03 | 98.01 | 96.40 | 94.70 | 81.31 | 72.31 | 74.61 |
| SimpleNet[15] | 93.42 | 91.16 | 87.27 | 98.50 | 96.91 | 95.40 | 82.17 | 69.82 | 73.59 |
| Ours | **97.51** | **95.27** | **93.35** | 99.60 | **98.53** | **97.80** | **90.15** | **81.12** | **86.01** |

Table 6: Ablation study on MVTec AD and APTOS. Reported in last/best

| | $\mathcal{L}_{global}$ | optim. encoder | stop grad. | contrast pairs | hard mining | MVTec AD | | APTOS |
|---|---|---|---|---|---|---|---|---|
| | | | | | | I-AUROC | AUPRO | I-AUROC |
| *Config.A* | | | | | | 95.31/97.55 | 93.34/94.05 | 90.12/90.50 |
| *Config.B* | ✓ | | | | | 98.86/99.07 | 94.51/94.59 | 92.49/93.62 |
| *Config.B+hm* | ✓ | | | | ✓ | 99.00/99.12 | 94.86/94.91 | 93.87/94.36 |
| *Config.C* | ✓ | ✓ | | | | 91.54/95.96 | 88.24/92.14 | 90.71/91.06 |
| *Config.D* | ✓ | ✓ | ✓ | | | 94.64/97.07 | 84.11/87.38 | 93.06/95.66 |
| *Config.C+cp* | ✓ | ✓ | | ✓ | | 97.59/97.88 | 93.76/93.92 | 92.68/92.68 |
| *Config.E−gl* | | ✓ | ✓ | ✓ | | 98.93/99.26 | 94.65/94.71 | 96.39/96.39 |
| *Config.E* | ✓ | ✓ | ✓ | ✓ | | 99.13/99.34 | 94.59/94.60 | 97.32/97.43 |
| Ours | ✓ | ✓ | ✓ | ✓ | ✓ | **99.45/99.52** | **95.20/95.29** | **97.51/97.51** |

# 4 Conclusion

In this study, we propose a novel contrastive learning paradigm, namely ReContrast, for domain-specific unsupervised anomaly detection. It addresses the transfer ability of the pre-trained encoder by jointly optimizing all parameters end-to-end. The key elements of contrastive learning are elegantly embedded in epistemic UAD method to avoid pattern collapse, training instability, and identical shortcut. Extensive experiments on MVTec AD, VisA, and three medical image datasets demonstrate our superiority. The idea of optimizing encoders can further boost the application of UAD methods on more image modalities that are far from natural image domain.

# 5 Limitations

**Scope of Application** In this work, we mainly focus on UAD that detects regional defects (most common in practical applications like industrial inspection and medical disease screening), which is distinguished from one-class classification (OCC, or Semantic AD). In our UAD, normal and anomalous samples are semantically the same objects except local detects, e.g. good cable v.s. spoiled cable. In OCC, normal samples and anomalous samples are semantically different, e.g. cat v.s. other animals. More details are discussed in Appendix E.

**Training Instability** Our method still suffers some extent of training instability. Because of the absence of validation sets in UAD settings, whether the last epoch (for reporting results) is in the middle of a loss spike and performance dip is related to random seeds. We found that a number of UAD methods (RD4AD[2], CFA[14], SimpleNet[15]) are also subject to training instability when running their code. We discuss the reason and solutions to mitigate this effect in Appendix E.

## Acknowledgments and Disclosure of Funding

This research work is supported by the National Natural Science Foundation of China (NSFC) (GrantNo. 82072007), Beijing Natural Science Foundation (Grant No. IS23112), and Beijing Institute of Technology Research Fund Program for Young Scholars.

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

# Appendices

First, we give the formulation of unsupervised anomaly detection (UAD). Let $\mathcal{D}^{train} = \{(x_i, y_i)\}_{i=1}^{N}$, $y_i \in \{-\}$ be the training set of normal samples, where $x_i$ is the $i^{th}$ sample with its label $y_i$. During test, $\mathcal{D}^{test} = \{(x_i, y_i)\}_{i=1}^{M}$, $y_i \in \{-, +\}$ ($-$ for normal, $+$ for anomalous) contains both normal and anomalous images. Our goal is to train a model using $\mathcal{D}^{train}$ to identify $(x, +)$ from $(x, -)$ in $\mathcal{D}^{test}$, while localizing anomalous regions at the same time.

## A    Related Work

**Epistemic methods** are based on the assumption that the networks respond differently during inference between seen input and unseen input. Within this paradigm, *pixel reconstruction* methods assume that the networks trained on normal images can reconstruct anomaly-free regions well, but poorly for anomalous regions [38]. Auto-encoder (AE) [39; 38], variational auto-encoder (VAE) [40; 41], or generative adversarial network (GAN) [42; 7; 8] are used to restore normal pixels. However, *pixel reconstruction* models may also succeed in restoring unseen anomalous regions if they resemble normal regions in pixel values or the anomalies are barely noticeable [2; 9]. Therefore, *feature reconstruction* is proposed to construct features of pre-trained encoders instead of raw pixels [2; 3; 4]. To prevent the whole network from converging to a trivial solution, the parameters of the encoders are frozen during training [9]. In *feature distillation* [2; 5; 6], the student network is trained from scratch to mimic the output features of the pre-trained teacher network with the same input of normal images, also based on the similar hypothesis that the student trained on normal samples only succeed in mimicking features of normal regions.

**Pseudo-anomaly** methods generate handcrafted defects on normal images to imitate anomalies, converting UAD to supervised classification [35] or segmentation task [36]. Specifically, CutPaste [35] simulates anomalous regions by randomly pasting cropped patches of normal images. DRAEM [36] constructs abnormal regions using Perlin noise as the mask and another image as the additive anomaly. SimpleNet [15] introduces anomaly by injecting Gaussian noise in the pre-trained feature space. These methods deeply rely on how well the pseudo anomalies match the real anomalies [2], which makes it hard to generalize to different datasets.

**Feature memory & modeling** methods [10; 11; 12; 13; 14] memorize all (or their modeled distribution) normal features extracted by networks pre-trained on large-scale datasets and match them with test samples during inference. Since these methods require memorizing, processing, and matching nearly all features from training samples, they are computationally expensive in both training and inference, especially when the training set is large. There is a considerable semantic gap between large-scale natural images on which the frozen networks were pre-trained and the target UAD images in various modalities. Therefore, methods based on pre-trained networks struggle in transferring as the feature encoders are not optimized in the target domain.

There are a few studies addressing the adaptation problem in UAD settings. CFA [14] and SimpleNet[15] unanimously utilize a learnable linear layer to transform the output features of the frozen encoder. However, the linear layer may be insufficient for adaption and can hardly recover the domain-specific information in pre-trained features, especially when the target domain, e.g. CT and MRI, is remote from natural image domain. In OCC settings, a number of works focus on utilizing self-supervised pre-training approaches to learn a compact representation space for normal samples [43; 44; 45]. However, the performances are far from satisfactory in UAD tasks (I-AUROC under 90% on MVTec AD).

## B    Datasets

**MVTec AD** [19] is an industrial defect detection dataset, containing 15 sub-datasets (5 texture categories and 10 object categories) with a total of 3,629 normal images as the training set and 1,725 normal and anomalous images as the test set. Pixel-level annotations are available for anomalous images for evaluating anomaly segmentation. All images are resized to 256$\times$256.

**VisA** [20] is an industrial defect detection dataset, containing 12 sub-datasets with a total of 9,621 normal and 1200 anomalous images. Pixel-level annotations are available for anomalous images for evaluating anomaly segmentation. We split it into training and test sets following the setting in [20]. All images are resized to $256\times256$.

**OCT2017** is an optical coherence tomography (OCT) dataset [21]. The dataset is labeled into four classes: normal, drusen, DME (diabetic macular edema), and CNV (choroidal neovascularization). The latter three classes are considered anomalous. The training set contains 26,315 normal images and the test set contains 1,000 images. All images are resized to $256\times256$.

**APTOS** is a color fundus image dataset, available as the official training set of the 2019 APTOS blindness detection challenge [22]. The dataset contains 3,662 images with annotated grade 0-4 to indicate different severity of diabetic retinopathy, yielding 1,805 normal images (grade 0) and 1857 anomalous images (grade 1-4). We randomly selected 1,000 normal images as our training set and the rest 2,662 images as the test set. Images are preprocessed to crop the fundus region, and then resized to $256\times256$.

**ISIC2018** is a skin disease dataset, available as task 3 of ISIC2018 challenge [23]. It contains seven classes. NV (nevus) is taken as the normal class and the rest of classes are taken as anomaly, following [32]. The training set contains 6705 normal images. The official validation set is used as our test set, which includes 193 images. Images are resized to $256\times256$ and then center cropped into $224\times224$ to remove redundant background. In addition, because the normal and anomalous images of ISIC2018 are different objects (lesions) instead of healthy objects and unhealthy objects, the task is more like one-class classification. Therefore, we take the mean (instead of maximum) value of $\mathcal{S}^{map}$ as the $\mathcal{S}^{img}$ in ReContrast and other reproduced methods.

## C   Complete Implementation Details

WideResNet50 [30] pre-trained on ImageNet [1] is utilized as the encoder by default. The decoder is the upside-down version of the encoder, exactly the same as RD4AD [2], i.e., the $3\times3$ convolution with stride 2 at the beginning of each layer is replaced by a $2\times2$ transposed convolution with stride 2. The bottleneck is also the same as RD4AD [2]. For each decoder layer, two 1x1 convolutions are used to project the feature map to reconstruct the frozen encoder and trained encoder, respectively. AdamW optimizer [31] is utilized with $\beta$=(0.9,0.999) and weight decay=1e-5. The learning rates of the new (decoder and bottleneck) and pre-trained (encoder) parameters are 2e-3 and 1e-5 respectively. The network is trained for 3,000 iterations on each sub-dataset in VisA, 2,000 on each sub-dataset in MVTec AD and ISIC2018, and 1,000 on APTOS and OCT2017. The batch size is 16 for industrial datasets and 32 for medical datasets. The $\alpha$ in (5) linearly rises from -3 to 1 in the first one-tenth iterations and keeps 1 for the rest of the training. By default, the batchnorm (BN) layers of encoder are set to *train* mode during training. Because training instability and performance drop are observed for some categories [3], the BN of encoder is set to *eval* mode (use pre-trained statistics) for such datasets. The reason is discussed in Appendix E. A library[4] is used to plot the landscape in Figure 4. The *distance* and *steps* arguments are set to 1 and 50 respectively.

On MVTec AD and VisA, the results of comparison methods are taken from the original papers or the benchmark paper [46] that report their performances. On medical datasets, we take the results from [32] if available; otherwise, we reproduce the method with official code. For RD4AD, we use their official code, in which $\mathcal{L}_{global}$ is already implemented instead of $\mathcal{L}_{region}$, as discussed in Section 2.1. Codes are implemented with Python 3.8 and PyTorch 1.12.0 cuda 11.3. Experiments are run on NVIDIA GeForce RTX3090 GPUs (24GB).

## D   Additional Experiments

### D.1   Additional Experimental Results

In addition to the averaged segmentation performances in Table 2, we present the P-AUROC and AUPRO of each categories in MVTec AD in Table A1. The results in the main paper are reported

---

[3]*Toothbrush*, *Leather*, *Grid*, *Tile*, *Wood*, *Screw* in MVTec AD, *cashew*, *pcb1* in VisA, and OCT2017.
[4]https://github.com/marcellodebernardi/loss-landscapes

with a single random seed following our baseline [2]. In Table A2, we report the mean and standard deviation of three runs with three different random seeds (1, 11, and 111). In addition to the averaged performances in Table 3, we present the I-AUROC, P-AUROC, and AUPRO of each category in VisA in Table A3.

Table A1: Anomaly segmentation performance of 15 subset (categories) of MVTec AD (%).

|         | Carpet | Grid | Leather | Tile | Wood | Bottle | Cable | Capsule |
|---------|--------|------|---------|------|------|--------|-------|---------|
| P-AUROC | 99.3   | 99.2 | 99.5    | 96.3 | 95.9 | 99.0   | 98.9  | 98.4    |
| AUPRO   | 97.9   | 97.8 | 99.2    | 93.7 | 92.5 | 97.1   | 95.6  | 95.4    |

|         | Hazelnut | MetalNut | Pill | Screw | Toothbrush | Transistor | Zipper |
|---------|----------|----------|------|-------|------------|------------|--------|
| P-AUROC | 99.1     | 98.7     | 99.1 | 99.6  | 99.2       | 95.4       | 98.1   |
| AUPRO   | 95.9     | 94.4     | 97.7 | 98.6  | 95.0       | 82.3       | 94.9   |

Table A2: Performance on MVTec AD over three runs (%).

|            | I-AUROC          | P-AUROC          | AUPRO            |
|------------|------------------|------------------|------------------|
| Carpet     | $99.72 \pm 0.27$ | $99.27 \pm 0.05$ | $98.55 \pm 0.88$ |
| Grid       | $100 \pm 0.00$   | $99.26 \pm 0.05$ | $97.79 \pm 0.03$ |
| Leather    | $100 \pm 0.00$   | $99.48 \pm 0.02$ | $99.17 \pm 0.04$ |
| Tile       | $99.66 \pm 0.17$ | $96.18 \pm 0.11$ | $92.95 \pm 0.66$ |
| Wood       | $99.05 \pm 0.05$ | $95.94 \pm 0.03$ | $92.61 \pm 0.10$ |
| Bottle     | $100 \pm 0.00$   | $99.00 \pm 0.01$ | $97.13 \pm 0.08$ |
| Cable      | $99.58 \pm 0.15$ | $98.92 \pm 0.02$ | $95.63 \pm 0.02$ |
| Capsule    | $97.86 \pm 0.36$ | $98.40 \pm 0.00$ | $95.34 \pm 0.04$ |
| Hazelnut   | $100 \pm 0.00$   | $99.08 \pm 0.03$ | $95.87 \pm 0.08$ |
| MetalNut   | $100 \pm 0.00$   | $98.73 \pm 0.04$ | $94.49 \pm 0.09$ |
| Pill       | $98.94 \pm 0.24$ | $99.13 \pm 0.03$ | $97.75 \pm 0.05$ |
| Screw      | $97.80 \pm 0.14$ | $99.57 \pm 0.02$ | $98.50 \pm 0.07$ |
| Toothbrush | $99.44 \pm 0.45$ | $99.17 \pm 0.02$ | $94.99 \pm 0.02$ |
| Transistor | $99.65 \pm 0.11$ | $95.38 \pm 0.03$ | $82.61 \pm 0.22$ |
| Zipper     | $99.68 \pm 0.15$ | $98.16 \pm 0.04$ | $95.12 \pm 0.25$ |
| *All Avg*  | $99.42 \pm 0.05$ | $98.39 \pm 0.01$ | $95.21 \pm 0.06$ |

Table A3: Performance of 12 subset (categories) of VisA (%).

|         | candle | capsules | cashew | chewinggum | fryum | macaroni1 |
|---------|--------|----------|--------|------------|-------|-----------|
| I-AUROC | 97.20  | 93.55    | 98.14  | 99.28      | 97.56 | 98.84     |
| P-AUROC | 99.15  | 99.46    | 97.41  | 97.36      | 91.96 | 99.01     |
| AUPRO   | 94.77  | 94.45    | 94.28  | 86.63      | 79.11 | 93.63     |

|         | macaroni2 | pcb1  | pcb2  | pcb3  | pcb4  | pipe_fryum |
|---------|-----------|-------|-------|-------|-------|------------|
| I-AUROC | 91.79     | 97.86 | 97.84 | 98.18 | 99.82 | 99.96      |
| P-AUROC | 99.04     | 99.79 | 98.99 | 99.05 | 98.70 | 98.30      |
| AUPRO   | 97.36     | 96.76 | 92.47 | 95.14 | 91.29 | 95.57      |

## D.2 Qualitative Visualization

The qualitative anomaly segmentation results of ReContrast trained by $\mathcal{L}_{global}$ and ReContrast trained by $\mathcal{L}_{global-hm}$ are presented in Figure A1. It is shown that the hard-normal regions are less activated with $\mathcal{L}_{global-hm}$. The results of VisA and medical images are shown in Figure A2 and Figure A3, respectively. Each $\mathcal{S}^{map}$ is min-max-normalized to 0-1 for clearer visualization.

## D.3 Additional Ablation Study

Ablation study is conducted on the value of $\alpha$ in $\mathcal{L}_{global-hm}$, as shown in Table A4. Assuming the regional cosine distances $\mathcal{M}^k(h, w)$ conforms Gaussian distribution, the discarding rates of feature

point are 2.3%, 15.9%, 50%, 69.1%, 84.1%, 93.3% and 97.7% for $\alpha$ = -2, -1, 0, 0.5, 1, 1.5 and 2, respectively. $\mathcal{L}_{global-hm}$ is not sensitive to the discarding rate within the range from 50% to 93%. Though $\alpha$ can be tuned for each dataset, we find $\alpha = 1$ works just well for most circumstances.

We test a variety of encoder backbones in Table A5, i.e., ResNet18, ResNet50, and WideResNet50 (default), and report their performances, model parameters, as well as multiply–accumulate operations (MACs). The corresponding decoder is the reversed version of the encoder. Different encoders may favor different training hyper-parameters such as learning rate and iteration. Though we do not further tune each backbone, all backbones produce excellent results with default hyper-parameters, suggesting the generality of our method. With each backbone, our method outperforms the corresponding feature reconstruction counterpart RD4AD [2]. Notably. our method with ResNet18 is comparable to RD4AD with WideResNet50.

The utilization of two encoders (one frozen, one trained domain-specific) in our method can function as an ensembling, which may also contribute to performances. We conduct ablation experiments using different encoder-decoder feature map pairs to generate anomaly maps in inference. The results are shown in Table A6. On MVTec, using only the feature of domain-specific encoder produces an I-AUROC of 99.38%, which is comparable to the 99.45% of the ReContrast default and outperforms the 98.86% of baseline Config. B. In addition, we train an ensembling version of Config. B using two different encoders (ResNet50 and WideResNet50), producing an I-AUROC of 98.94% which is nearly identical to the baseline. On APTOS, using only the feature of domain-specific encoder produces an I-AUROC of 97.73%, which outperforms both 97.51% of the ReContrast default and 92.49% of the baseline Config. B. The results indicate that the improvement due to ensembling is small and the domain-specific encoder is vital.

Table A4: Ablation on the values of $\alpha$ in $\mathcal{L}_{global-hm}$ on MVTec AD (%).

| $\alpha$ | -inf ($\mathcal{L}_{global}$) | -2 | -1 | 0 | 0.5 | 1 | 1.5 | 2 |
|---|---|---|---|---|---|---|---|---|
| discard rate | 0% | 2.3% | 15.9% | 50% | 69.1% | 84.1% | 93.3% | 97.7% |
| I-AUROC | 99.13 | 99.13 | 99.14 | 99.32 | 99.39 | **99.45** | 99.38 | 98.85 |
| P-AUROC | 98.09 | 98.11 | 98.13 | 98.30 | 98.31 | 98.37 | **98.39** | 98.25 |
| AUPRO | 94.59 | 94.62 | 94.65 | 94.97 | 95.00 | 95.20 | **95.28** | 95.13 |

Table A5: Ablation on the encoder backbones on MVTec AD (%).

| Backbone | ResNet18 | | ResNet50 | | WResNet50 | |
|---|---|---|---|---|---|---|
| Method | RD4AD | Ours | RD4AD | Ours | RD4AD | Ours |
| Params(M) | 18.7 | 21.7 | 63.6 | 74.9 | 117 | 145 |
| MACs(G) | 4.95 | 10.1 | 19.4 | 42.0 | 36.0 | 75.2 |
| I-AUROC | 97.9 | **98.7** | 98.4 | **99.1** | 98.5 | **99.5** |
| P-AUROC | 97.1 | **97.9** | 97.7 | **98.3** | 97.8 | **98.4** |
| AUPRO | 91.2 | **94.2** | 93.1 | **95.0** | 93.9 | **95.2** |

Table A6: Ablation on the use of different encoder feature maps for calculating $\mathcal{S}^{map}$ (%).

| | encoder feature maps for calculating $\mathcal{S}^{map}$ | MVTec AD | APTOS |
|---|---|---|---|
| Config. B | 3 frozen | 98.86 | 92.49 |
| Config. B (R50+WR50) | 6 frozen | 98.94 | 92.14 |
| Ours | 3 frozen only | 99.16 | 95.32 |
| Ours | 3 trained only | 99.38 | **97.73** |
| Ours | 3 trained + 3 frozen | **99.45** | 97.51 |

# E   Limitations

**Scope of Application.** In this work, we mainly focus on UAD that detects regional defects (most common in practical applications), which is distinguished from one-class classification (OCC, or

Semantic AD). In our UAD, normal and anomalous samples are semantically the same objects except local detects, e.g. good cable v.s. spoiled cable. In OCC, normal samples and anomalous samples are semantically different, e.g. cat v.s. other animals. The slight difference in task setting makes the focus of corresponding methods different. Most methods for UAD attempt to detect anomalies by segmenting anomalous local regions, while methods for OCC detect anomalies based on the representation deviation of the whole image [47; 48]. Though methods of these two tasks can be used for each other to a certain extent, we mainly focus on UAD in this work.

**Logical Anomaly**. MVTec LOCO is a recently released dataset for evaluating the detection performance of logical anomalies. It contains 5 sub-datasets, each comprised of both structural anomalies, e.g. dents and scratches as in MVTec AD, and logical anomalies, e.g. dislocation and missing parts. We find our method performing less favorably on logical anomalous images than GCAD [49] which is specially designed for logical anomalies, as presented in Table A7. We still outperform other vanilla UAD methods that are not specially designed for such anomalies.

Table A7: Anomaly Detection Performance on MVTec LOCO (%). Structural I-AUROC is calculated on normal images and structural anomalous images. Logical I-AUROC is calculated on normal images and logical anomalous images. Mean I-AUROC is the average of the above two scores.

|  | RD4AD [2] | DRAEM [36] | PaDiM [10] | PatchCore [11] | GCAD [49] | Ours |
|---|---|---|---|---|---|---|
| struct. I-AUROC | 88.0 | 74.4 | 70.5 | 82.0 | 80.6 | **90.7** |
| logic. I-AUROC | 69.4 | 72.8 | 63.7 | 69.0 | **86.0** | 73.4 |
| mean I-AUROC | 78.7 | 73.6 | 67.1 | 75.5 | **83.3** | 82.1 |

**Batch Normalization and Training Instability**. As discussed in Appendix C, training instability and performance drops are observed for a few categories when setting BN mode of the encoder to *train*. The phenomenon of training instability is observed in many deep learning tasks. However, validation sets are often allowed, which counteracts training instability and helps the choice of BN mode.

On the one hand, it can be caused by the limited feature diversity of the one-class UAD dataset. First, we recall the formulation of BN layer. Giving a $d$-dimensional feature $x = (x^{(1)}, ..., x^{(d)})$, each dimension (channel) is normalized in training:

$$\hat{x}^{(k)} = \frac{x^{(k)} - E\left[x^{(k)}\right]}{\sqrt{Var\left[x^{(k)}\right] + \epsilon}}$$

where the expectation $E$ and variance $Var$ are computed over the batch, and $\epsilon$ is a small constant for numerical stability. In most computer vision datasets, an input batch contains a variety of categories. Each dimension is more or less activated by different image features so that $Var\left[x^{(k)}\right]$ is not zero. However, because a UAD dataset contains only one type of category, there is a chance that a channel of pre-trained feature $x^{(k)}$ is barely activated or equally activated, e.g. a channel sensitive to animals is dead on industrial objects. Thus, the denominator $\sqrt{Var\left[x^{(k)}\right] + \epsilon}$ is nearly zero, causing malfunction of batch normalization. This problem can be fixed by using pre-trained statistical *running_mean* and *running_var* (*eval* mode), which loses the effect of BN.

On the other hand, this instability can be attributed to the feature of Adam optimizer. We found that a number of UAD methods [2; 14; 15] are also subject to training instability when running their code. Whether the last epoch (for reporting results) is in the middle of a loss spike and performance dip is strongly related to random seed. In some works [50; 51], the authors attribute the training instability to the historical estimation of squared gradients in Adam optimizer.

In our according explorations, we replace the batch variance $Var\left[x^{(k)}\right]$ smaller than min(5e-4, *running_var*) by pre-trained *running_var* and we reset the historical state buffer (gradient momentum and second-order gradient momentum) of Adam every 500 iters. Without manually selecting BN mode, such tricks enable training with more stability. It yields I-AUROC of 99.41% on MVTec and 97.28% on VisA, comparable to the 99.45% and 97.5% of our reported result and still outperforms previous SOTAs. In the future, approaches to eliminating the problem of BN and optimizer can be further investigated.

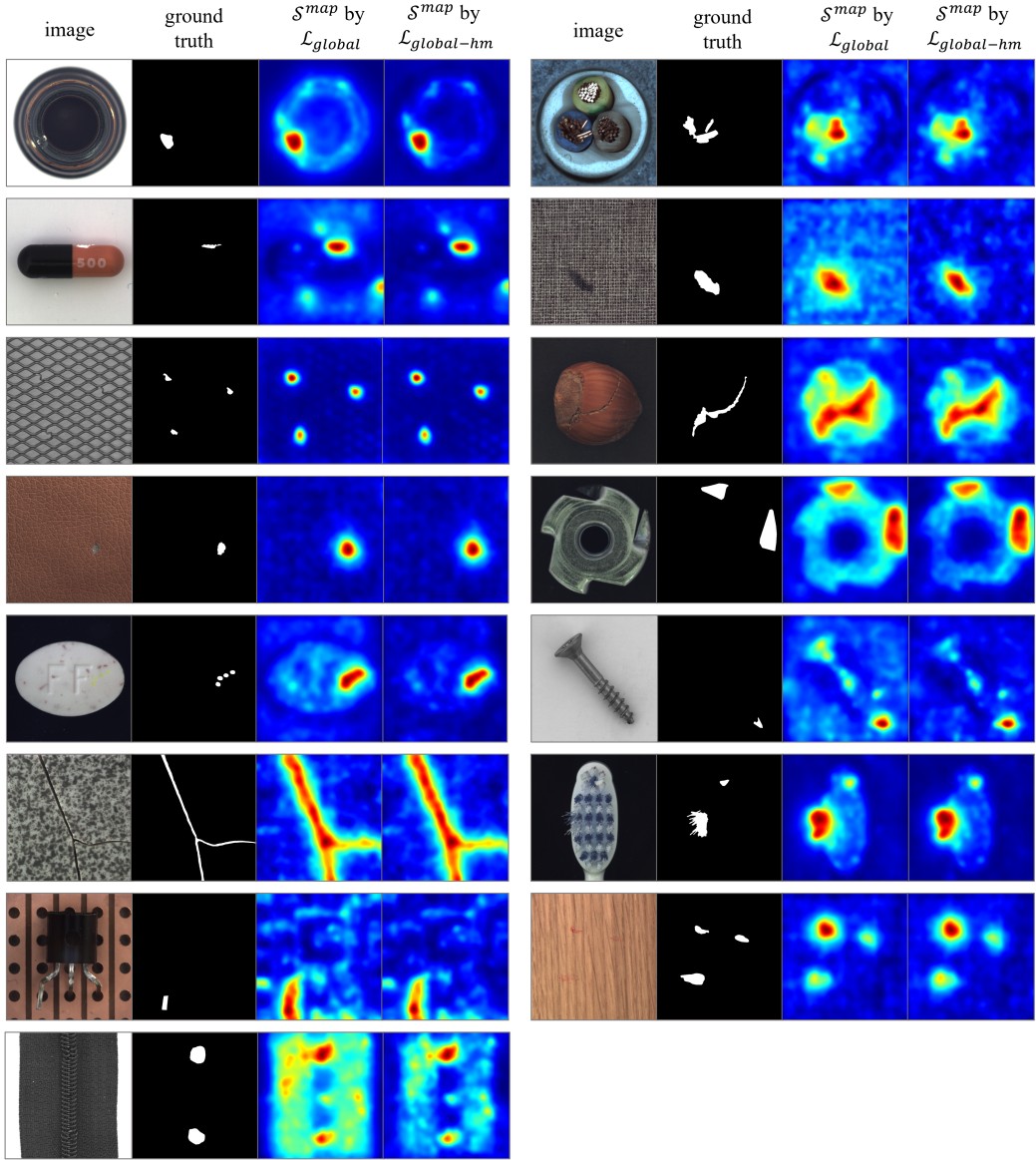

Figure A1: $\mathcal{S}^{map}$ of our ReContrast trained by $\mathcal{L}_{global}$ or $\mathcal{L}_{global-hm}$. Hard-normal regions are less activated with $\mathcal{L}_{global-hm}$.

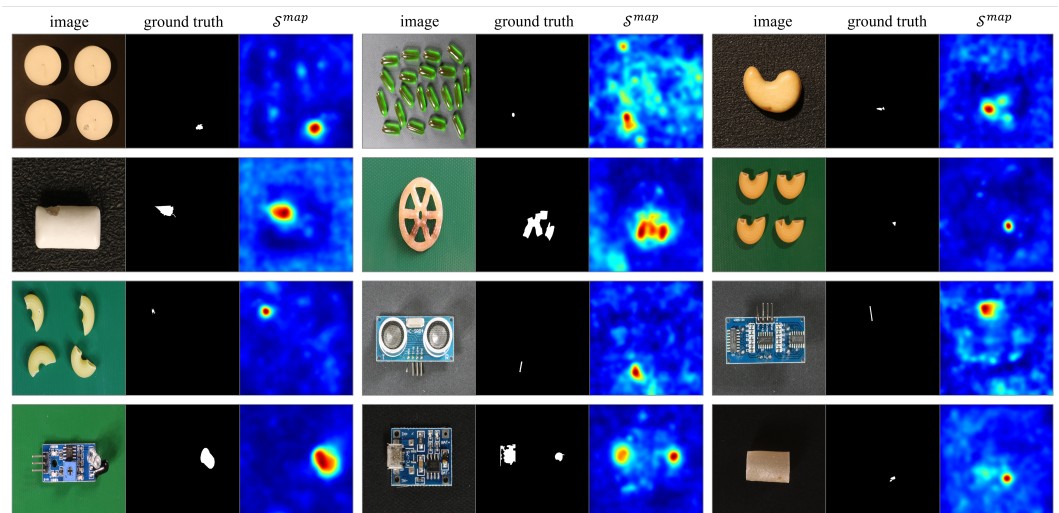

Figure A2: $\mathcal{S}^{map}$ of our method on VisA.

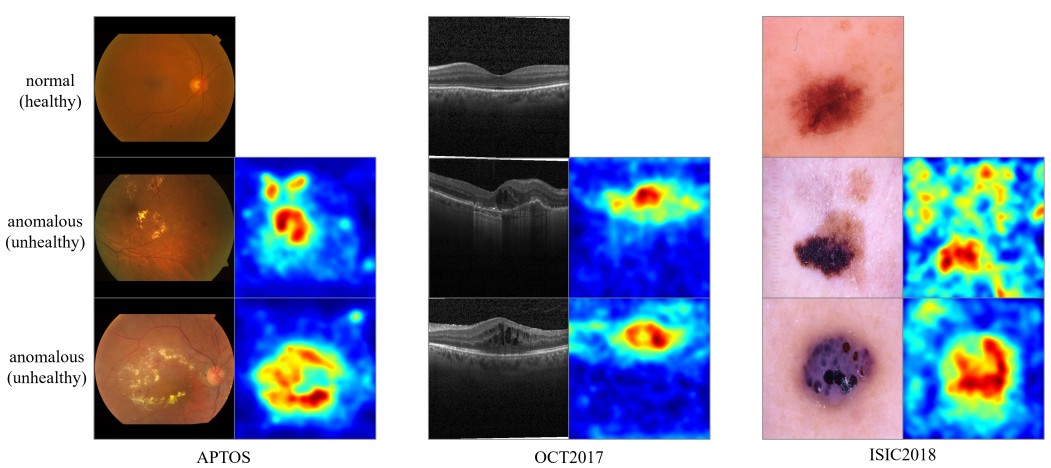

Figure A3: $\mathcal{S}^{map}$ of our method on medical image datasets.

