# OpenReview forum: "ReContrast: Domain-Specific Anomaly Detection via Contrastive Reconstruction"
_NeurIPS.cc/2023/Conference — NeurIPS 2023 poster_

### Official Review · Reviewer_S3Cv · 2023-06-20

**Soundness:** 3 good
**Presentation:** 3 good
**Contribution:** 2 fair
**Rating:** 6
**Confidence:** 4

**Summary:**

The paper presents an approach, called ReContrast, for domain-specific anomaly detection and segmentation. The authors aim to overcome the transfer limitations of existing methods by addressing and mitigating the challenges associated with the use of pre-trained encoders from natural image domains. They propose a combination of the reverse distillation approach and contrastive learning to optimize the entire network and align it with the target domain.

In the paper, the authors systematically explain their approach, highlighting three key elements: the introduction of a new optimization objective called global cosine distance to enhance training stability, the utilization of stop-gradient operation to prevent pattern collapse, and the integration of contrastive learning (without the use of negative samples). By integrating these elements into their feature reconstruction framework, the authors improve the network's transfer ability for anomaly detection tasks.


**Strengths:**

1. The authors address and analyze the limitations of the reverse distillation approach by incorporating simple yet effective modifications. This results in an improved anomaly detection method.
2. The paper is well-written, and provides a clear and concise description of the proposed ReContrast framework and its underlying principles.
3. The experimental evaluations conducted across diverse datasets and models demonstrate the practicality and effectiveness of the approach.


**Weaknesses:**

1. The authors handle training instability by selectively changing the batch normalization (BN) mode of the encoder based on performance on specific datasets or categories. This raises concerns about the proposed method's reliability and generalizability. Using the test set to modify the training process of problematic categories can introduce unintended biases and compromise the evaluation integrity. It would be more desirable to provide a systematic and principled approach to utilizing BN layers without relying on a separate validation set. This would ensure consistent and unbiased training across all datasets and categories. Consequently, it is important to report the performance of the proposed approach when utilizing the train and eval modes consistently for all datasets. Such analysis would provide valuable insights into the observed performance drops that the authors mentioned and help assess the robustness of the proposed method.

2. Another limitation of the paper is the narrow focus on structural anomalies, as evidenced by the results presented in Table A6. While the proposed method performs well at detecting structural anomalies, it falls short when it comes to detecting logical anomalies. It is important to highlight this limitation in the main paper since unsupervised anomaly detection encompasses various anomaly types, including global anomalies like those found in datasets such as CIFAR-10 ([24-26] in the appendix). Clarifying the scope and limitations of the method for detecting different anomaly types will provide a more comprehensive understanding of its applicability in real-world scenarios.

\#### Update #### \
I've increased my rating slightly in light of the response.

**Questions:**

The paper lacks of reported results in the multi-class setting, where anomaly detection is performed across multiple target classes simultaneously. This setting ([9] in the manuscript), reflects a more realistic scenario of anomaly detection in real-world applications. By learning the data distribution of multiple classes jointly, the detection task becomes more complex and challenging. However, the paper does not provide insights into the performance of the proposed ReContrast approach in this specific setting. What is the performance of the proposed ReContrast approach in this setting?

I am willing to improve my score once these points (including weaknesses) are clarified.

**Limitations:**

Yes.

---

> ### Author Rebuttal · Authors · 2023-08-09
>
> Thank you for your insightful comments and suggestions.
>
> __1. The authors handle training instability by selectively changing the batch normalization (BN) mode of the encoder based on performance on specific datasets or categories. This raises concerns about the proposed method's reliability and generalizability.__
>
> Thanks for this valuable question. This instability issue actually lies in the optimization of neural networks.
>
> As discussed in Appendix E line 134-144, one possible reason is the denominator of BN layer: “Because a UAD dataset contains only one type of category, there is a chance that a channel is barely activated or equally activated, driving the variance of BN to zero.”. We have preliminarily shown that we can reach a comparable result 99.33% on MVTecAD by replacing the batch variance smaller than a constant to pre-trained running_var for each channel.
>
> In addition, we found that a number of training-based UAD methods (RD4AD[2], CFA[14], SimpleNet[15]) also suffer from training instability when running their code. Whether the last epoch (for reporting results) is in the middle of a loss spike and performance dip is strongly related to random seed. The phenomenon of training instability is also observed in many other disciplines. However, validation sets are allowed which mitigate the effect of instability. In [a], the authors attribute the training instability on large-scale datasets to the historical gradient estimation of Adam optimizer. Accordingly, we tried a strategy to reset the running state (momentum and velocity) of Adam every 1000 iterations. Without manually selecting BN mode, it yields I-AUROC of 99.37% on MVTec and 97.3% on VisA, which is comparable to the 99.45% and 97.5% of our reported result and still outperforms previous SOTAs.
>
> Such approach, analysis, and results will be added into the main paper for fair comparison and to provide insights for training UAD models under potential instability.
>
> [a] Molybog, Igor, et al. "A theory on adam instability in large-scale machine learning." arXiv preprint arXiv:2304.09871 (2023).
>
> __2. Another limitation of the paper is the narrow focus on structural anomalies. It is important to highlight this limitation in the main paper since unsupervised anomaly detection encompasses various anomaly types, including logical anomalies and global anomalies like those found in datasets such as CIFAR-10.__
>
> Thank you for this insightful comment.
>
> Due to the page limitation of the manuscript, we discuss the field of UAD in Appendix A lines 7-14. We distinguish UAD from one-class classification (OCC, or Semantic AD). In our UAD, normal samples and anomalous samples are semantically the objects except local detects, e.g. good cable v.s. spoiled cable, healthy eye v.s. diseased eye. In OCC, normal samples and anomalous samples are semantically in different categories, e.g. cat v.s. other animals (like in CIFAR-10). The slight difference in task setting makes the focus of corresponding methods different. Most methods for UAD attempt to detect anomalies by segmenting anomalous regions, while methods for OCC detect anomalies based on the representation deviation of the whole image. The cross-using of methods in different settings causes unsatisfactory performances because they address different problems with different anomaly assumptions, as discussed in Appendix lines 48-51. In this work, we mainly focus on UAD that detects regional defects (most common in industrial inspection and medical disease screening).  We will add these claims of scope at the beginning of the Method section in the main paper for clarity.
>
> __3. The unified setting (all-class-one-model) of UniAD, reflects a more realistic scenario of anomaly detection. What is the performance of the proposed ReContrast approach in this setting?__
>
> Thank you for your valuable suggestion. We run ReContrast under the unified setting on MVTec-AD and VisA. The model is optimized on the whole dataset for 5000 iterations. On MVTec AD, our method produces an I-AUROC of __98.2%__, exceeding the previous SOTA 96.5% of UniAD [9] by a large 1.7%. The results on VisA are presented in global reply due to page limitation. The results under the unified setting will be added in the final version following your valuable suggestion.
>
> __I-AUROC (%) on MVTec AD under unified setting__
> |     |  |   |   |
> |:----------:|:-----:|:-----:|:----------:|
> | Method     | RD4AD | UniAD | ReContrast |
> | Carpet     | 98.3  | __99.8__  | 98.3 |
> | Grid       | __99.0__  | 98.2  | 98.9 |
> | Leather    | __100__   | __100__   | __100__ |
> | Tile       | 98.1  | 99.3  | __99.5__       |
> | Wood       | 99.3  | 98.6  | __99.7__       |
> | Bottle     | 79.9  | 99.7  | __100__        |
> | Cable      | 86.8  | 95.2  | __95.6__       |
> | Capsule    | 96.3  | 86.9  | __97.3__       |
> | Hazelnut   | __100__   | 99.8  | __100__        |
> | MetalNut   | 99.8  | 99.2  | __100__        |
> | Pill       | 92.8  | 93.7  | __96.3__       |
> | Screw      | 96.5  | 87.5  | __97.2__       |
> | Toothbrush | __97.5__  | 94.2  | 96.7       |
> | Transistor | 93.3  | __99.8__  | 94.5       |
> | Zipper     | 98.6  | 95.8  | __99.4__ |
> | Mean       | 95.8  | 96.5  | __98.2__       |
>
>
> The reason why we did not include such results in the initial submission is that we consider this setting is actually less practical than the conventional separate setting (one-class-one-model). For example, in medical images, it is unreasonable to train a unified model for all modalities including CT, MRI, fundus images, etc, when the categories of images are easy to obtain. Similarly in industrial defect detection, the categories of the object are determined for each production line. The main benefit of a unified model is to save disk memory for model parameters, which is not the focus of this paper.

---

> > ### Comment · Reviewer_S3Cv · 2023-08-14
> >
> > Thank you for your thorough response. I have increased my rating accordingly.

---

### Official Review · Reviewer_e2Vq · 2023-06-23

**Soundness:** 3 good
**Presentation:** 2 fair
**Contribution:** 2 fair
**Rating:** 4
**Confidence:** 5

**Summary:**

This paper proposes an epistemic method, namely ReContrast, for unsupervised anomaly detection (UAD). The proposed method is deeply based on UniAD and RD4AD, with some technical improvement to adapt to the data from different domains. Experiments are conducted on two popular industrial defect detection benchmarks and three medical image UAD tasks.

**Strengths:**

1. The technical details are well illustrated in the paper.
2. Ablation studies show the effectiveness of the proposed components.

**Weaknesses:**

1.	The paper lacks novelty and resembles more of a technical report than a research paper. The authors summarize their contributions in Lines 68-78, which primarily consist of detailed technical modifications. The proposed method heavily relies on existing methods UniAD and RD4AD, with some specific improvements such as generating contrastive pairs and utilizing a GAP-mimicking objective function.
2.	This paper has room for improvement in terms of its writing. In Lines 42-59, the authors introduce several components aimed at enhancing performance. However, the relationships between these components are not clearly established, and the motivations behind their inclusion are not adequately explained.


**Questions:**

1.	Unclear contributions compared to existing methods: The paper does not clearly differentiate its contributions from those of existing methods. Both feature reconstruction [9] and contrastive learning [a,b] have been extensively explored in the anomaly detection literature. Although the authors mention the usage of stop-gradient as their main contribution (Lines 75-76), it has already been employed in [a,b].
2.	Missing important experimental comparisons: While this paper heavily relies on UniAD [9], it fails to provide any comparisons with UniAD in Table 1, Table 2, and Table 3. Including such comparisons would be crucial for evaluating the proposed method against the baseline.
3.	Poor organization: The organization of Section 2.2-2.6 is confusing and lacks coherence. These subsections appear to be assembled randomly without a clear structure, and there is no apparent connection between them. It seems that existing methods and the authors' proposed methods are randomly mixed together, leading to a lack of clarity and logical flow.
4.	Domain-specific claims and evaluation: The authors assert that their method is domain-specific, claiming that the proposed components could help adapt the model to industrial or medical domains. However, it is unclear whether the trained model can be used simultaneously on both industrial datasets, i.e, MVTec and Visa. The authors should clarify why they separately evaluate these datasets. They are suggested to follow the approach of UniAD [9] to demonstrate if one model can effectively detect anomalies across multiple categories within a single dataset.

References:
[a] Focus your distribution: Coarse-to-fine non-contrastive learning for anomaly detection and localization. ICME 2022.
[b] Registration-based Few-Shot Anomaly Detection. ECCV 2022.

---

> ### Author Rebuttal · Authors · 2023-08-09
>
> Thank you for all your valuable suggestion and concerns.
>
> __1. (Weakness 1 & Question 1) The paper lacks novelty. The proposed method is deeply based on UniAD and RD4AD, with some technical improvement to adapt to the data from different domains. Unclear contributions compared to existing methods [a,b]__
>
> First, we would like to say that our method is not based on UniAD, where transformer decoder, neighbor masking, and feature jitter are proposed to address the unified UAD setting (all-class-one-model). The only thing in common is that they are both feature-reconstruction-based, which is a big paradigm of UAD. It is fair to say that our method is based on RD4AD, which serves as our baseline and starting point of exploration.
>
> Though [a] and [b] utilize stop-grad of contrastive learning (CL) to avoid pattern collapse, they are not contrastive methods (nor did the authors state they are), they did not analyze how stop-grad helps, and their motivation is not to optimize pre-trained encoder. In contrast, we further dive into how stop-grad helps to prevent pattern collapse by probing the feature diversity of the encoder, as discussed in Sec.2.3 and Sec.2.4. In addition, stop-grad is only one component of CL, but ReContrast introduces three cohesive elements of CL (global similarity, stop-grad, and contrastive pairs) to build a complete 2-D contrastive paradigm. In the final version, we will cite [a] and [b] to review the use of stop-grad in UAD tasks.
>
> Second, we would like to claim that it is not the simple assembling of existing techniques that makes our method specific to a specific dataset. It is the motivation to optimize and reorient the feature encoder towards each target UAD training set that makes our method domain-specific to whatever it is trained on, instead of using a universal ImageNet-pretrained frozen encoder even if it is not discriminative on the target dataset.
>
> Under this global motivation, we found it difficult to optimize the pre-trained encoder during feature reconstruction training accompanied by pattern collapse, instability, and identical shortcut, which is why all previous methods freeze the pre-trained encoder without any adaption during training. Inspired by the structural similarity between CL and feature reconstruction, we bring three essences of CL into feature reconstruction to enable smooth optimization of encoder. We would like to say that the three proposed components, i.e., global cosine distance, stop gradient, and augmentation-free contrastive pair, are NOT A+B+C-like combinations, but three cohesive components strongly correlated and inspired by CL. In addition, except stop-grad, the elements of CL cannot be directly utilized in feature reconstruction UAD. For example, because the global representation after global average pooling (GAP) in CL messes locality, we propose a novel global cosine distance that calculates the distance in 1-D while still maintaining the pixel-to-pixel correspondence. Because the image augmentation of CL cannot be used in UAD scenario, we propose an augmentation-free method to construct two views of one image.
>
> __2. (Weakness 2 & Question 3) This paper has room for improvement in terms of its writing. The paper resembles more of a technical report than a research paper.__
>
> Inspired by the narrative of a number of previous arts (SimSiam, MoCo v3), we tried a writing logic of “spot a challenge, connection to CL, address the problem, analysis the mechanism”, instead of the conventional manner of “methods in Method, experiments in Experiment”. This writing logic involves bringing some preliminary experiments in the Method section and analysis of training dynamics (feature diversity, loss landscape). We believe such an exploration-like narrative illustrates the motivation and intuition of each proposed component better.
>
> To be specific, in Introduction, we point out that the three proposed components are cohesively inspired by the elements of CL, which is the reason we name it ReContrast. In the Method section, we found it too intuitive and confusing to directly propose three components without explaining why to introduce and how they work. Therefore, in each section, we first present an obstacle of optimizing encoder in feature reconstruction, which is the motivation of the proposed components. Second, we target an element of CL that address a similar challenge in representation learning. Third, we adapt the elements to fit in the 2-D feature reconstruction paradigm. And last, we analyze the mechanism by inspecting feature diversity or loss-parameter landscape.
>
> Furthermore, some reviewers (especially #3 xS4v) acknowledge this narrative, and we will be very grateful if you feel comfortable with this paper structure. In the final version, we will further emphasize each method’s cohesiveness, motivation, and specific relation to contrastive learning inspired by your valuable comments.
>
> __3. (Question 2) While this paper heavily relies on UniAD [9], it fails to provide any comparisons.__
>
> Since our method is not based on UniAD (please refer to the reply of Question 1) and not evaluated under unified setting, we compared the separate setting (one-class-one-model) version of UniAD, i.e. ADTR [3]. ADTR is the work of the same authors of UniAD with exactly the same architecture but without the components addressing the unified setting (all-class-one-model), which is harmful to separate setting. ADTR (I-AUROC=97.5%) is better than UniAD (96.6%) under the separate setting, which we believe is much fairer to be compared with our method.
>
> __4. (Question 4) They are suggested to follow the unified-setting of UniAD [9] (all-class-one-model).__
>
> __Because of the character limitation of each response, we place the response in the global reply. Please check.__

---

> > ### Comment · Reviewer_e2Vq · 2023-08-17
> >
> > Thank you for your reply. I have carefully read the rebuttal and other reviews. As the authors have addressed part of my concerns, I have slightly increased my score.

---

### Official Review · Reviewer_xS4v · 2023-07-01

**Soundness:** 3 good
**Presentation:** 3 good
**Contribution:** 3 good
**Rating:** 7
**Confidence:** 5

**Summary:**

This paper proposes a novel unsupervised anomaly detection (UAD) method called ReContrast. The method addresses the issue of poor transferability of pre-trained encoders from natural image domains to target UAD domains such as industrial inspection and medical imaging. ReContrast optimizes the entire network to reduce biases towards the pre-trained image domain and aligns the network with the target domain. It combines feature reconstruction and contrastive learning to prevent training instability and pattern collapse. The method is evaluated on industrial defect detection benchmarks and medical image UAD tasks, demonstrating its superiority over current state-of-the-art methods.

**Strengths:**

1. The paper is well-written and easy to read.
2. This paper propose a new unsupervised anomaly detection method, ReContrast. According to the suggested setup, it is technically valid and should work well.
3. Sufficient experiments are done to demonstrate the effectiveness of the method, and the robustness of the method is proved by experiments on multiple data sets
4. The results were strong. Performance is excellent in most cases.

**Weaknesses:**

1. In the ablation experiments of MVTec AD and APTOS, for example, the experiment on Global Cosine Similarity, the author conducted experiments on the original RD4AD, namely Config A and Config B. This is not sufficient to prove the validity of Global Cosine Similarity on ReContrast, and may be a special case on RD4AD Settings.
2. The realistic contribution of this paper is relatively weak. Compared with other methods, the reconstructed method does not rely on the pre-trained model or the memory library in anomaly detection. The method in this paper relies on the pre-trained model to maintain the feature extraction capability of the encoder, which I think may affect the practical application of the model.

**Questions:**

I really appreciate the structure of this article. The author has done a lot of work in the demonstration experiment, but the method innovation and contribution are relatively limited, but the overall is worth learning. Here are some suggestions.
1.Regarding the experiment of Global Cosine Similarity, the author conducted the experiment on the original RD4AD, namely Config A and Config B. It is suggested to add the experimental setting using Region Cosine Similarity to the experimental setting of Config E. To demonstrate the validity of Global Cosine Similarity on ReContrast, rather than a special case on RD4AD Settings.
2.In the experimental part, I noticed that the P-AUROC/AUPRO index lacked two model indexes compared with I-AUROC, ADTR and CFlow AD, in the experiment conducted on MVTec AD dataset. I hope the author could provide completion to ensure the completeness of the experiment.
3.It is hoped that the author can visualize how the model gets the Anomaly score map in the visual RD4AD in Figure 2 to ReContrast, so as to facilitate readers' understanding.

**Limitations:**

Limitations are clearly described and I don't expect negative societal impact of this work.

---

> ### Author Rebuttal · Authors · 2023-08-09
>
> First and foremost, we are thrilled that you like our writing logic. We made a lot of considerations to use this exploration-like narrative instead of a conventional paper structure.
>
> __1. In the ablation experiments of MVTec AD and APTOS, for example, the experiment on Global Cosine Similarity, the author conducted experiments on the original RD4AD, namely Config A and Config B. This is not sufficient to prove the validity of Global Cosine Similarity on ReContrast, and may be a special case on RD4AD Settings.__
>
> Thank you for your valuable suggestion. We have conducted an ablation experiment replacing the global cosine loss of Config.E by regional cosine loss, called Config.E-R. The results are shown in the following table.  On MVTec AD, Config. E-R produces a final I-AUROC of 98.93%. On APTOS, Config. E-R produces a final I-AUROC of 96.39%. The performance is inferior to Config.E with global cosine but superior to Config.A without other proposed contrastive elements, which demonstrates the validity of global cosine loss as well as stop-grad and contrastive pairs. The results will be added to Table 5 in the final version.
>
> __I-AUROC (%) on MVTec AD and APTOS.__
> | | | | | | | |
> |------|:------------:|:-----------------:|:-------------:|:------------------:|:------------:|:-----------:|
> | **-**       | **$L_{global}$** | **optim-encoder** | **stop-grad** | **contrast-pairs** | **MVTec AD** | **APTOS**   |
> | Config. A   |              |                   |               |                    | 95.31  | 90.12 |
> | Config. B   | √             |                   |               |                    | 98.86  | 92.49 |
> | Config. E-R |              | √                 | √             | √                  | 98.93  | 96.39 |
> | Config. E   | √            | √                 | √             | √                  | 99.13  | 97.32 |
>
> __2.The realistic contribution of this paper is relatively weak. Compared with other methods, the reconstructed method does not rely on the pre-trained model or the memory library in anomaly detection. The method in this paper relies on the pre-trained model to maintain the feature extraction capability of the encoder, which I think may affect the practical application of the model.__
>
> Thank you for this comment. The reconstructed methods you refer to are the image-reconstruction methods that train Auto-Encoder-like and GAN-like networks to reconstruct raw pixels from scratch. In line 21-22 of Appendix, we review such methods and point out their insufficiency: “However, pixel reconstruction models may also succeed in restoring unseen anomalous regions if they resemble normal regions in pixel values or the anomalies are barely noticeable [10; 11].” As far as we know, the best approach in this paradigm is RIAD [a] which produces an I-AUROC of only 91.7% on MVTec AD, compared with the 96%+ of recent methods that utilize pre-trained encoders. In addition, we also compare two such methods (f-AnoGAN, GANomaly) on medical datasets in Table 4, which also produces undesirable results.
>
> We totally agree with your comments on “methods relied on pre-trained model limits the application”. It is actually the motivation of our work. In line 34-35 of the manuscript, we mentioned that “In previous works, poor transfer ability is caused by the semantic gap between natural images on which the frozen networks were pre-trained and various UAD image modalities”. The main motivation of this paper is to reorient the large-scale pre-trained encoders to UAD datasets including medical images that are far from ImageNet, e.g. CT and MRI, which promotes the practical applications of UAD on various image domains and modalities.
>
> [a] Zavrtanik, V., Kristan, M., & Skočaj, D. (2021). Reconstruction by inpainting for visual anomaly detection. Pattern Recognition, 112, 107706.
>
> __3. In the experimental part, I noticed that the P-AUROC/AUPRO index lacked two model indexes compared with I-AUROC, ADTR and CFlow AD, in the experiment conducted on MVTec AD dataset.__
>
> Due to the limitation of page length and width of the initial submission, some comparison results are not put in. The I-AUROC and AUPRO of CFlow-AD are 98.62% and 94.60%. The I-AUROC of ADTR is 96.1%. We will add the results in the Appendix in final version.
>
> __4. It is hoped that the author can visualize how the model gets the Anomaly score map in the visual RD4AD in Figure 2 to ReContrast, so as to facilitate readers' understanding.__
>
> Thank you for your valuable suggestion. We add the visualization of score map calculation in Figure 2, as shown in the PDF in the global reply.

---

> > ### Comment · Reviewer_xS4v · 2023-08-17
> >
> > I appreciate the authors for their responses to my concerns. In summary, I will raise my score from weak accept to accept, in expectation of your inclusion of some discussion in the final version

---

### Official Review · Reviewer_dCnY · 2023-07-05

**Soundness:** 2 fair
**Presentation:** 3 good
**Contribution:** 2 fair
**Rating:** 7
**Confidence:** 5

**Summary:**

This work proposes an unsupervised anomaly detection method using contrastive learning method to reduce the biases between the pre-trained image domain and the target domain. In the paper, different designs of reconstruction networks are introduced and compared in terms of detection performance, model coverage and feature diversity. Finally, authors adopt a contrastive learning method equipped with global cosine loss, hard example mining, as well as additional frozen pre-trained encoder branch. This work conducts extensive experiments across two industrial defect detection benchmarks and three medical image anomaly detection tasks to demonstrate the transfer ability on various image domains.

**Strengths:**

+ This work analyzes the domain gap between the pretrained image domain and the dataset image domain and tries a variety of modifications on distillation network to reduce the domain gap.
+ Extensive experiments are conducted on MVTec AD, VisA, and three medical image datasets to demonstrate the transfer ability on various image domains.

**Weaknesses:**

- During both training and inference, the feature loss is calculated using six feature pairs from two encoders. The utilization of two encoders in the methods leads to an increase in parameters and inference time. However, this approach functions as an ensemble, contributing to improved performance. It is important to note that the ablation study does not provide results solely generated by the domain-specific encoder.
- The training process for configuration E lacks explicit description in the text. In accordance with the information provided in Figure 2, the encoders are switched and repeated during training. However, the text does not provide a detailed explanation of this process.
- Figure 4 displays the landscapes of S_map, which allows for a comparison between the two losses. According to the description, S_map represents the average anomaly score map of the APTOS dataset. However, the paper or reference [27] does not provide an introduction on how to interpret or visualize the figures, nor does it explain why average anomaly score maps from different images are used, considering that they may have different anomaly locations.

**Questions:**

- As the work follows the SimSiam’s stop-gradient operation to prevent model collapsing. How about using momentum encoder in Moco to train the domain-specific encoder?
- In this framework, the encoder must be switched during training, why not merge the features from flexible encoder and frozen encoder and then fed to the neck and decoder? In this way a unified network can be achieved.

**Limitations:**

Authors have addressed limitations of the method on training instability and logical anomaly detection.

---

> ### Author Rebuttal · Authors · 2023-08-09
>
> We would like to thank you for your constructive comments with great insights.
>
> __1. During both training and inference, the feature loss is calculated using six feature pairs from two encoders. The utilization of two encoders in the methods leads to an increase in parameters and inference time. However, this approach functions as an ensemble, contributing to improved performance. It is important to note that the ablation study does not provide results solely generated by the domain-specific encoder.__
>
> Thank you for the insightful suggestions. Following your advice, we have conducted ablation experiments using the domain-specific encoder only to generate anomaly maps in inference. The results are shown in the following table. On MVTec, using only the feature of domain-specific encoder produces an I-AUROC of 99.38%, which is comparable to the 99.45% of the ReContrast default and outperforms the 98.86% of baseline Config. B. In addition, we train an ensembled version of Config. B using two different encoders (ResNet50 and WRN50), producing an I-AUROC of 98.94% which is nearly identical to the baseline. On APTOS, using only the feature of domain-specific encoder produces an I-AUROC of 97.73%, which outperforms both 97.51% of the ReContrast default and 92.49% of the baseline Config. B. The results indicate that the improvement due to ensembling is small and the domain-specific encoder is vital. These ablation experiments will be added in the final version.
>
> __I-AUROC (%) on MVTec AD and APTOS.__
> |   |    |     |  |
> |:--------------------:|:----------------------------:|:------------:|:---------:|
> | **-**                | **encoder feature of $S_{map}$** | **MVTec AD** | **APTOS** |
> | Ours (default)        | 3 trained+3 frozen           | 99.45        | 97.51     |
> | Ours        | 3 trained only   | 99.38        | 97.73     |
> | Ours                 | 3 frozen only     | 99.16        | 95.32     |
> | Config.B             | 3 frozen      | 98.86        | 92.49     |
> | Config. B (R50+WR50) | 6 frozen   | 98.94        | 92.14     |
>
> __2. The training process for configuration E lacks explicit description in the text. In accordance with the information provided in Figure 2, the encoders are switched and repeated during training. However, the text does not provide a detailed explanation of this process.__
>
> Thank you for your valuable comments. The “switch and repeat” means that the decoder takes the representation from the trainable encoder to reconstruct the pre-trained encoder, while it also takes the representation from the pre-trained encoder to reconstruct the trainable encoder at the same time (switch position). In the manuscript, it is described in line 190-192: “The decoder and bottleneck take the features of one encoder as the input to reconstruct the features of the other encoder and __vice versa__.”
>
> In the final version, we will make a clearer description following your suggestion: “The decoder and bottleneck take the features of the domain-specific encoder to reconstruct the frozen encoder; vice versa, they simultaneously take the features of the frozen encoder to reconstruct the domain-specific encoder, building a complete double-view contrastive paradigm.” We will revise Figure.1 and Figure.2-Config. E to be more precise, as shown in the PDF in the global reply.
>
> __3. Figure 4 displays the landscapes of S_map. According to the description, S_map represents the average anomaly score map of the APTOS dataset. However, the paper or reference [27] does not provide an introduction on how to interpret or visualize the figures, nor does it explain why average anomaly score maps from different images are used, considering that they may have different anomaly locations.__
>
> Thank you for your comments and we will add descriptions in the final version to make it clearer to readers. The x-y plane (horizontal) does not represent the location in images. Given a network architecture and its trained parameters, this tool [27] calculates and visualizes the loss (or metric, i.e. averaged S_map) surface along random directions near the given (optimal) model parameters. Therefore, the figures visualize the averaged S_map (a scaler, average of all locations of all samples, you can consider it as mean L_region) on z-axis, against the model parameter space (reduced to 2-dimension) on x-y plane (not image locations). This metric v.s. parameter landscape indicates the stability of training and the generalization ability. Axis labels are added in the revised Figure 4 as presented in the PDF in global reply.
>
> __4. As the work follows the SimSiam’s stop-gradient operation to prevent model collapsing. How about using momentum encoder in Moco to train the domain-specific encoder?__
>
> Thank you for this insightful question. We have tried to use an EMA (exponential moving average momentum) version of the domain-specific encoder to substitute the frozen encoder, which makes it similar to BYOL [17]. It yields an I-AUROC of 99.43% on MVTec, which is comparable to the 99.45% of our default ReContrast. This result also complies with the conclusion of SimSiam that it is the implicit stop-grad operation in the momentum encoder that prevents the collapsing  .
>
> __5. In this framework, the encoder must be switched during training, why not merge the features from flexible encoder and frozen encoder and then fed to the neck and decoder? In this way a unified network can be achieved.__
>
> Thank you for this valuable comment. The “switch and repeat” represents the simultaneous cross-reconstruction between two encoders. Therefore, if you mean merging the features of flexible encoder and frozen encoder on batch dimension, it is exactly what we mean and do. We concatenate two groups of features in a batch and then feed them to the neck and decoder. Then we split the output of decoder back into two groups. We have revised the architecture of Config. E in Figure 2 for clearer illustration, as shown in the PDF in the global reply.

---

> > ### Comment · Reviewer_dCnY · 2023-08-20
> >
> > Thank you for authors' reply. The rebuttal addressed most of my concerns. I slightly increased my score.

---

### Official Review · Reviewer_Jpdb · 2023-07-11

**Soundness:** 3 good
**Presentation:** 3 good
**Contribution:** 2 fair
**Rating:** 4
**Confidence:** 5

**Summary:**

This paper introduces a feature reconstruction model, namely ReContrast, for anomaly detection. By incorporating a set of techniques including feature global similarity, stop gradient, contrastive pair optimizations, the authors shows good performance of ReContrast on both industrial and medical datasets.

**Strengths:**

1. The paper is easy to follow. The logic behind some technical improvements is well motivated.

2. Extensive experimentation on diverse anomaly detection datasets is conducted. The numerical results of ReContrast outperforms prior arts.

3. The analysis of limitation of RD4AD, especially the Global Cosine Similarity, is impressive.

**Weaknesses:**

1. The paper presentation should be improved. The current version is in a lab report format, outlining the observed issues and employing trial-and-error strategies to address them. It would be preferable to present the methodology in a more cohesive manner.

2. I find it unclear why Config. D would prevent model collapse. As the feature similarity loss can be back-propagated from the decoder to the encoder, the encoder and decoder may collaborate to achieve a trivial solution.

3. The motivation behind Config. E is also unclear to me. If the decoder's objective is to reconstruct the pre-trained encoder, what is the purpose of the trainable encoder? What is the objective function in Config. E?

4. Figure 4 illustrates the model landscape with respect to model's trainable parameters. A flatter surface indicates a more stable training process, but it does not necessarily indicate better generalization concerning input data.

**Questions:**

Please refer to question 2-4 in the weakness section.

**Limitations:**

The appendix discuses the limitations of this method.

---

> ### Author Rebuttal · Authors · 2023-08-04
>
> We would like to thank you for your constructive comments with great insights. The replies to specific remarks are as followed:
>
> __1. The paper presentation should be improved. The current version is in a lab report format, outlining the observed issues and employing trial-and-error strategies to address them. It would be preferable to present the methodology in a more cohesive manner.__
>
> Thank you for this kind suggestion. Inspired by the narrative of a number of previous arts [a, b], we tried a writing style of “spot a challenge, analysis the mechanism, solve the problem”, instead of the conventional manner of “methods in Method, experiments in Experiment”. This writing logic involves bringing some primary experiments in the Method section. We believe such an exploration-like narrative illustrates the motivation and intuition of our proposed method better. Though three challenges and solutions (global cosine, stop-grad, contrastive pair) are presented sequentially, they are all inspired by and strongly correlated to a cohesive center, i.e., elements of contrastive representation learning.
>
>
> __2. I find it unclear why Config. D would prevent model collapse. As the feature similarity loss can be back-propagated from the decoder to the encoder, the encoder and decoder may collaborate to achieve a trivial solution.__
>
> A short and intuitive explanation of stop-gradient operation, which is wildly used to prevent model collapse in contrastive representation learning (CL) [a, c], is that the optimization goals of the encoder and decoder (a.k.a predictor in CL) are different though gradients are back-propagated together. The goal of the decoder is to reconstruct non-constant encoder features, while the goal of the encoder is to provide discriminative information so that the decoder can reconstruct non-constant informative objections. As long as the encoder is not initialized to generate constant features, the configuration will not collapse to a trivial solution. We call this explanation “mutual reinforcement” in line 169-171.
>
> There are also a number of works that try to explain the deeper mechanism of stop gradient. In [a], the authors hypothesize this configuration as an implementation of an Expectation-Maximization (EM) algorithm that implicitly solves two underlying sub-problems with two sets of variables. In [d], the authors argue that there are flaws in the hypothesis of [a] and suggest that the decomposed gradient vector (center gradient) helps prevent collapse via the de-centering effect. We will refer to them as further explanations of the mechanism of stop-grad in the final version.
>
>
> __3. The motivation behind Config. E is also unclear to me. If the decoder's objective is to reconstruct the pre-trained encoder, what is the purpose of the trainable encoder? What is the objective function in Config. E?__
>
> We apologize for the ambiguity caused. The decoder takes the representation from the trainable encoder to reconstruct the pre-trained encoder, while it also takes the representation from the pre-trained encoder to reconstruct the trainable encoder __at the same time__.
>
> The objective function of Config. E is $L_{global}(E_{train}, D_{freeze})+ L_{global}(E_{freeze}, D_{train})$, where $E_{train}$ is the feature from trainable encoder, $E_{freeze}$ is the feature from frozen encoder,  $D_{freeze}$ is the decoder feature with the input of frozen encoder, $D_{train}$ is the decoder feature with the input of trainable encoder.
>
> In the final version, we will make a clearer description in Sec.2.5 as: “The decoder and bottleneck take the features of the domain-specific encoder to reconstruct the frozen encoder; vice versa, they simultaneously take the features of the frozen encoder to reconstruct the domain-specific encoder, building a complete two-view contrastive paradigm.” We revise Figure.1 and Figure.2 to be more precise, as shown in the PDF in the global reply.
>
>
> __4. Figure 4 illustrates the model landscape with respect to model's trainable parameters. A flatter surface indicates a more stable training process, but it does not necessarily indicate better generalization concerning input data.__
>
> Previous works of literature [25, 26] suggest that the model in flat minima generalizes better than the model in sharp minima, including the well-known Sharp Awareness Minimization (SAM) [26] that implements this theory to design optimizer. (Please note that it is the flatness of the area near the optimized local minimum, not the whole landscape). The generalization is actually unwanted in UAD settings because the model may also well reconstruct unseen anomalies. We refer to this theory to explain why global cosine performs better than regional cosine after converging to a sharper local minimum. We will make a clearer claim in the final version that it is the sharpness around local minima not global landscape that indicates the generalization ability.
>
> [a] X. Chen and K. He. “Exploring simple Siamese representation learning”. In Proceedings of the IEEE Computer Society Conference on Computer Vision and Pattern Recognition, 2021. ([16] in paper)
>
> [b] X. Chen, S. Xie, and K. He, “An Empirical Study of Training Self-Supervised Vision Transformers,” in Proceedings of the IEEE International Conference on Computer Vision, 2021.
>
> [c] J. B. Grill et al., “Bootstrap your own latent a new approach to self-supervised learning,” in
> Advances in Neural Information Processing Systems, 2020. ([17] in paper)
>
> [d] C Zhang et al., “How does simsiam avoid collapse without negative samples? a unified understanding with self-supervised contrastive learning”, in International Conference on Learning Representations, 2022.

---

> > ### Comment · Reviewer_Jpdb · 2023-08-19
> > **Thanks for the feedback from the authors.**
> >
> > The authors answered most of my questions. But I am still concerning the landscape figure. So I am keeping my original score.

---

> > > ### Author Response · Authors · 2023-08-21
> > > **On the connection between loss landscape and generalization**
> > >
> > > Thank you for your comments. The connection between the geometry of the loss landscape —— in particular, the flatness of minima —— and generalization has been studied from theoretical, empirical, and practical perspectives [a,b,c,d,e].
> > >
> > > A direct and intuitive explanation is given as followed. Because of the distribution deviation between train samples and unseen test samples, flat and wide minimum on training samples can wrap the minimum of test samples, while sharper minimum on training samples will differ more from the minimum of test samples which causes higher test loss. Please see Figure 1 of [a] for a clearer illustration, as we cannot upload figure here.
> > >
> > > [a] Keskar, N.S., Mudigere, D., Nocedal, J., Smelyanskiy, M. and Tang, P.T.P., 2016, November. On Large-Batch Training for Deep Learning: Generalization Gap and Sharp Minima. In International Conference on Learning Representations.
> > >
> > > [b] Dziugaite, G.K. and Roy, D.M., 2017. Computing nonvacuous generalization bounds for deep (stochastic) neural networks with many more parameters than training data. arXiv preprint arXiv:1703.11008.
> > >
> > > [c] Jiang, Y., Neyshabur, B., Mobahi, H., Krishnan, D. and Bengio, S., 2020. Fantastic generalization measures and where to find them. In International Conference on Learning Representations.
> > >
> > > [d]  Zhuang, J., Gong, B., Yuan, L., Cui, Y., Adam, H., Dvornek, N.C., s Duncan, J. and Liu, T., 2021, October. Surrogate Gap Minimization Improves Sharpness-Aware Training. In International Conference on Learning Representations.
> > >
> > > [e] Foret, P., Kleiner, A., Mobahi, H. and Neyshabur, B., 2020, October. Sharpness-aware Minimization for Efficiently Improving Generalization. In International Conference on Learning Representations.

---

### Author Rebuttal · Authors · 2023-08-09

__Note: *Revised Figures are in the one-page PDF. Please check "pdf" below.*__


__*The following area is only the extension of response to Reviewer 4 (e2VQ) due to page limitation.*__

__Reply to Review 4 (e2VQ). 4. (Question 4) They are suggested to follow the unified setting of UniAD [9] (all-class-one-model).__

We run our ReContrast under the unified setting on MVTec AD and VisA. The model is optimized on each whole dataset for 5000 iterations. On MVTec AD, our method produces an I-AUROC of __98.2%__, exceeding the previous SOTA 96.5% of UniAD by a large 1.7%. On VisA, both our ReContrast (__95.1%__) and baseline RD4AD (92.7%) outperform the previous SOTA method UniAD (88.8%), while the proposed method exceeds RD4AD by 2.4%. Detailed results on each category are presented in the following table. The results under this unified setting will be added in the final version following your constructive suggestion.


__I-AUROC (%) on MVTec AD under unified setting__
|     |  |   |   |
|:----------:|:-----:|:-----:|:----------:|
| Method     | RD4AD | UniAD | ReContrast |
| Carpet     | 98.3  | __99.8__  | 98.3 |
| Grid       | __99.0__  | 98.2  | 98.9 |
| Leather    | __100__   | __100__   | __100__ |
| Tile       | 98.1  | 99.3  | __99.5__       |
| Wood       | 99.3  | 98.6  | __99.7__       |
| Bottle     | 79.9  | 99.7  | __100__        |
| Cable      | 86.8  | 95.2  | __95.6__       |
| Capsule    | 96.3  | 86.9  | __97.3__       |
| Hazelnut   | __100__   | 99.8  | __100__        |
| MetalNut   | 99.8  | 99.2  | __100__        |
| Pill       | 92.8  | 93.7  | __96.3__       |
| Screw      | 96.5  | 87.5  | __97.2__       |
| Toothbrush | __97.5__  | 94.2  | 96.7       |
| Transistor | 93.3  | __99.8__  | 94.5       |
| Zipper     | 98.6  | 95.8  | __99.4__ |
| Mean       | 95.8  | 96.5  | __98.2__       |

__I-AUROC (%) on VisA under unified setting__
|           |        |       |            |
|:---------:|:------:|:-----:|:----------:|
| Method    | RD4AD  | UniAD | ReContrast |
| candle    | 93.6   | 94.0    | __96.3__    |
| capsules  | 67.8   | 72.2  | __77.7__       |
| cashew    | __94.6__   | 94.4  | 94.5       |
| chewingum | 95.3   | 98.1  | __98.6__       |
| fryum     | 94.4   | 86.3  | __97.3__       |
| macaroni1 | 97.2   | 88.5  | __97.6__       |
| macaroni2 | 86.2   | 75.3  | __89.5__       |
| pcb1      | __96.5__   | 90.1  | __96.5__       |
| pcb2      | 93.0     | 88.4  | __96.8__       |
| pcb3      | 94.5   | 84.1  | __96.8__       |
| pcb4      | __100__    | 99.0    | 99.9       |
| pip_fryum | __99.7__   | 94.7  | 99.3       |
| Mean      | 92.7   | 88.8  | __95.1__       |


The reason why we did not include such setting in the initial submission is that we consider this setting is actually less practical than the conventional separate setting. For example, in medical images, it is unreasonable to train a unified model for all modalities including CT, MRI, and fundus images, when the categories of images are easy to obtain. Similarly in industrial defect detection, the categories of the object are determined for each production line. The main benefit of a unified model is to save disk memory for model parameters, which is not the focus of this paper.

---

### Decision · Program_Chairs · 2023-09-21

**Decision:**

Accept (poster)

**Comment:**

This paper presents a novel contrastive learning paradigm, namely ReContrast, that aims to improve domain-specific unsupervised anomaly detection. It helps to overcome challenges such as pattern collapse, training instability, and the use of identical shortcuts that can hamper the learning process. The effectiveness of this approach has been validated through extensive experiments on various datasets, including MVTec AD, VisA, and three medical image datasets, showing superior performance compared to existing methods.

Initially, there were several criticisms raised by all reviewers, due to the ambiguity and uncertainty in the results. It also affected the reviewers' confidence in the validity of the results. The rebuttal could address many of the issues and after careful consideration and discussions after the rebuttal, the AC finds merits in the paper. If accepted, the authors are encouraged to address all issues raised by the reviewers.